# Integrated profiling of human pancreatic cancer organoids reveals chromatin accessibility features associated with drug sensitivity

Xiaohan Shi [1,10], Yunguang Li[2,3,10], Qiuyue Yuan [4,5,10], Shijie Tang [2,3,10], Shiwei Guo [1,10], Yehan Zhang[2,3], Juan He [2,3], Xiaoyu Zhang[2,3], Ming Han[2,3], Zhuang Liu[2,3], Yiqin Zhu[2], Suizhi Gao[1], Huan Wang [1], Xiongfei Xu [1], Kailian Zheng[1], Wei Jing[1], Luonan Chen [2,6,7,8 ✉], Yong Wang [4,5,6,7 ✉], Gang Jin [1 ✉] & Dong Gao [2,9 ✉]

Chromatin accessibility plays an essential role in controlling cellular identity and the therapeutic response of human cancers. However, the chromatin accessibility landscape and gene regulatory network of pancreatic cancer are largely uncharacterized. Here, we integrate the chromatin accessibility profiles of 84 pancreatic cancer organoid lines with whole-genome sequencing data, transcriptomic sequencing data and the results of drug sensitivity analysis of 283 epigenetic-related chemicals and 5 chemotherapeutic drugs. We identify distinct transcription factors that distinguish molecular subtypes of pancreatic cancer, predict numerous chromatin accessibility peaks associated with gene regulatory networks, discover regulatory noncoding mutations with potential as cancer drivers, and reveal the chromatin accessibility signatures associated with drug sensitivity. These results not only provide the chromatin accessibility atlas of pancreatic cancer but also suggest a systematic approach to comprehensively understand the gene regulatory network of pancreatic cancer in order to advance diagnosis and potential personalized medicine applications.

[1] Department of Hepatobiliary Pancreatic Surgery, Changhai Hospital, Second Military Medical University (Naval Medical University), Shanghai, China. [2] State Key Laboratory of Cell Biology, Shanghai Key Laboratory of Molecular Andrology, Shanghai Institute of Biochemistry and Cell Biology, Center for Excellence in Molecular Cell Science, Chinese Academy of Sciences, Shanghai 200031, China. [3] University of Chinese Academy of Sciences, Beijing 100049, China. [4] CEMS, NCMIS, HCMS, MDIS, Academy of Mathematics and Systems Science, Chinese Academy of Sciences, Beijing 100080, China. [5] School of Mathematical Sciences, University of Chinese Academy of Sciences, Beijing 100049, China. [6] Center for Excellence in Animal Evolution and Genetics, Chinese Academy of Sciences, Kunming 650223, China. [7] Key Laboratory of Systems Biology, Hangzhou Institute for Advanced Study, University of Chinese Academy of Sciences, Chinese Academy of Sciences, Hangzhou, China. [8] Guangdong Institute of Intelligence Science and Technology, Hengqin, Zhuhai, Guangdong 519031, China. [9] Institute for Stem Cell and Regeneration, Chinese Academy of Sciences, Beijing 100101, China. [10] These authors contributed equally: Xiaohan Shi, Yunguang Li, Qiuyue Yuan, Shijie Tang, Shiwei Guo. ✉email: lnchen@sibcb.ac.cn; ywang@amss.ac.cn; jingang@smmu.edu.cn; dong.gao@sibcb.ac.cn

Pancreatic cancer is one of the most aggressive malignancies, with a 5-year survival rate of 9%[1]. In the past decade, substantial advances have been made in the genetic and transcriptomic classification of pancreatic cancer[2–8]. However, therapeutic selection for pancreatic cancer patients is commonly based on clinical experience and patient status and is rarely predicted by the defined genetic and transcriptomic classifications. A detailed understanding of the molecular networks that integrate the landscapes of chromatin accessibility, DNA mutation, gene expression and drug sensitivity may provide insights into the therapeutic refractoriness of pancreatic cancer.

Recently, the application of the assay for transposase-accessible chromatin using the sequencing (ATAC-seq) method effectively enabled genome-wide profiling of chromatin accessibility with small sample quantities[9]. Chromatin accessibility landscapes has provided tremendous insights into the regulatory function to bridge the noncoding mutations and potential therapeutic strategies. Several previous studies indicated the critical effect of chromatin accessibility on identifying WNT signaling amplification in pancreatic cancer[10] and revealing an enhancer reprogramming-mediated developmental transition to metastasis[11]. However, the chromatin accessibility landscape of human pancreatic cancer has not been investigated and neglected in pan-cancer study[12].

The emerging patient-derived cancer organoid system could be a functional preclinical research platform for understanding pancreatic cancer progression and drug resistance[8,13–18]. Pancreatic ductal adenocarcinoma (PDAC) organoids have been generated to study patient chemosensitivity and develop personalized drug screening approaches[8,19]. Notably, three organoid lines derived from pancreatic neuroendocrine neoplasms (NENs) were recently generated[20]. Multiomics characterization of a large repertoire of patient-derived pancreatic cancer organoid (PDPCO) lines would improve the understanding and facilitate the development of new diagnostic and therapeutic strategies for this lethal disease.

In this work, we establish a PDPCO biobank including 4 NEN lines and thoroughly characterize these organoid lines by whole-genome sequencing (WGS), RNA sequencing (RNA-seq), ATAC-seq and drug sensitivity screening. We integrate the multiomics profiling data for PDPCOs, reveal functional pancreatic cancer-related noncoding mutations, provide new insights into the gene regulatory networks of different pancreatic cancer subtypes, and reveal a chromatin accessibility signature associated with drug sensitivity.

## Results

### Establishment and histological characterization of the PDPCOs. 
We seeded pancreatic cancer samples freshly collected by surgery or endoscopic ultrasound-guided fine-needle aspiration biopsy (EUS-FNA) into pancreatic cancer organoid culture medium and generated a total of 84 PDPCOs from 99 patients with exocrine pancreatic tumors (80.80%, 80 lines from 99 patient samples) and 39 patients with neuroendocrine tumors (10.26%, 4 lines from 39 patient samples) (Fig. 1a and Supplementary Data 1). In addition, 6 normal pancreatic ductal organoids (NPOs) were established. The pathological types of the original tumors were as follows: 75 PDACs, 4 intraductal papillary mucinous neoplasms (IPMNs), 1 acinar cell carcinoma (ACC) and 4 pancreatic NENs.

Although the PDPCOs showed great morphological diversity (Fig. 1b), they displayed the histological and immunohistological patterns present in the original patient tumor samples. Organoids derived from PDACs formed typical glandular tubular structures, as seen in the corresponding patient specimens (Fig. 1c and Supplementary Fig. 1a), and had similar expression levels of the ductal cell marker CK19 and the cell proliferation marker Ki67 (for example, CAS-DAC-21 and CAS-DAC-25) (Supplementary Fig. 1b). The mucinous neoplasm characteristic of IPMNs was maintained in CAS-IPMN-1, as evidenced by alcian blue staining (Fig. 1d). In CAS-ACC-1, the typical acinar cell markers α1-Antichymotrypsin (α1-ACT) and Bcl-10 were highly expressed, whereas the ductal cell marker CK19 and the neuroendocrine cell markers Chromogranin-A (CHGA) were not expressed (Supplementary Fig. 1c). Moreover, tumors generated by engrafting CAS-DACs, CAS-IPMN-1 and CAS-ACC-1 into SCID mice were reminiscent of the original tumor tissues (Fig. 1c, d and Supplementary Fig. 1a-c).

NEN is a rare but clinically important subtype of pancreatic neoplasia[21]. The efficiency of establishing continuously propagated organoid lines from NEN samples was approximately 10% (4 lines from 39 "attempts" with samples). CAS-NEN-1 and CAS-NEN-4 (derived from neuroendocrine carcinomas (NECs), with >50% positive Ki67 staining), CAS-NEN-2 (grade G2, with 3 to 20% positive Ki67 staining) and CAS-NEN-3 (grade G1, with <3% positive Ki67 staining), originating from pancreatic NENs, showed extensive neuroendocrine features, including islet morphology and positivity for Synaptophysin (SYP) and CHGA staining both in organoid culture and when engrafted in vivo (Fig. 1e and Supplementary Fig. 1d).

These results strongly indicated that the histological organization, differentiation status and morphological heterogeneity of the primary tumors were conserved in the PDPCOs.

### PDPCOs retain typical genomic alterations in pancreatic cancer. 
Chromosomal copy number variations (CNVs) constitute a common class of mutations that are an important mechanism of gene disruption or gene activation in pancreatic carcinogenesis[6,22]. To characterize the genomic alterations in the established PDPCOs, WGS was performed on 70 exocrine and 4 neuroendocrine tumor organoid lines, and the data were compared to those for the matched normal DNA from peripheral blood samples. The CNV segments of the exocrine PDPCOs showed high concordance with the TCGA pancreatic adenocarcinoma (PAAD) dataset[23], including the amplification of regions near KRAS, MYC and AKT2 and loss of regions near CDKN2A, TP53 and SMAD4 (Supplementary Fig. 2a). Strikingly, focal amplification of KRAS and increased inactivation events for CDKN2A, TP53 and SMAD4 were identified previously[6].

In total, 3,968 nonsynonymous somatic mutations were detected within coding regions (Supplementary Data 2). The 70 exocrine organoids harbored a median of 53.0 somatic mutations (range: 28–113) (Fig. 2), similar to that observed in the TCGA cohort[23]. The top mutated genes reported in the list of cancer-related genes in the COSMIC database[24] and OncoKB database[25], such as KRAS (96%), TP53 (86%), SMAD4 (37%), CDKN2A (24%), RNF43 (10%), ARID1A (10%), TGFBR2 (9%) and KMT2C (7%), were identified in PDPCOs (Fig. 2), consistent with the TCGA dataset[23](Supplementary Fig. 2b). $KRAS^{G12D}$ (48%, 32/67) was the most frequent KRAS mutation, while $KRAS^{G12V}$ (34%, 23/67), $KRAS^{G12R}$ (9%, 6/67) and $KRAS^{G12C}$ (3%, 2/67) were also identified in the exocrine organoid lines.

Three mutational signatures (A, B and C) were identified by the nonnegative matrix factorization (NMF) method in the 70 exocrine organoids, which were highly similar to the Cosmic signatures: Cosmic signature 5, Cosmic signature 17, and Cosmic signature 1, respectively (Supplementary Fig. 2c). Cosmic signature 1 and Cosmic signature 5 were reported to exist in all cancer types, and Cosmic signature 17 was also found in many gastrointestinal cancers. The aetiology of Cosmic signature 5 and 17 remained unknown, while Cosmic signature 1 was thought to be the result of

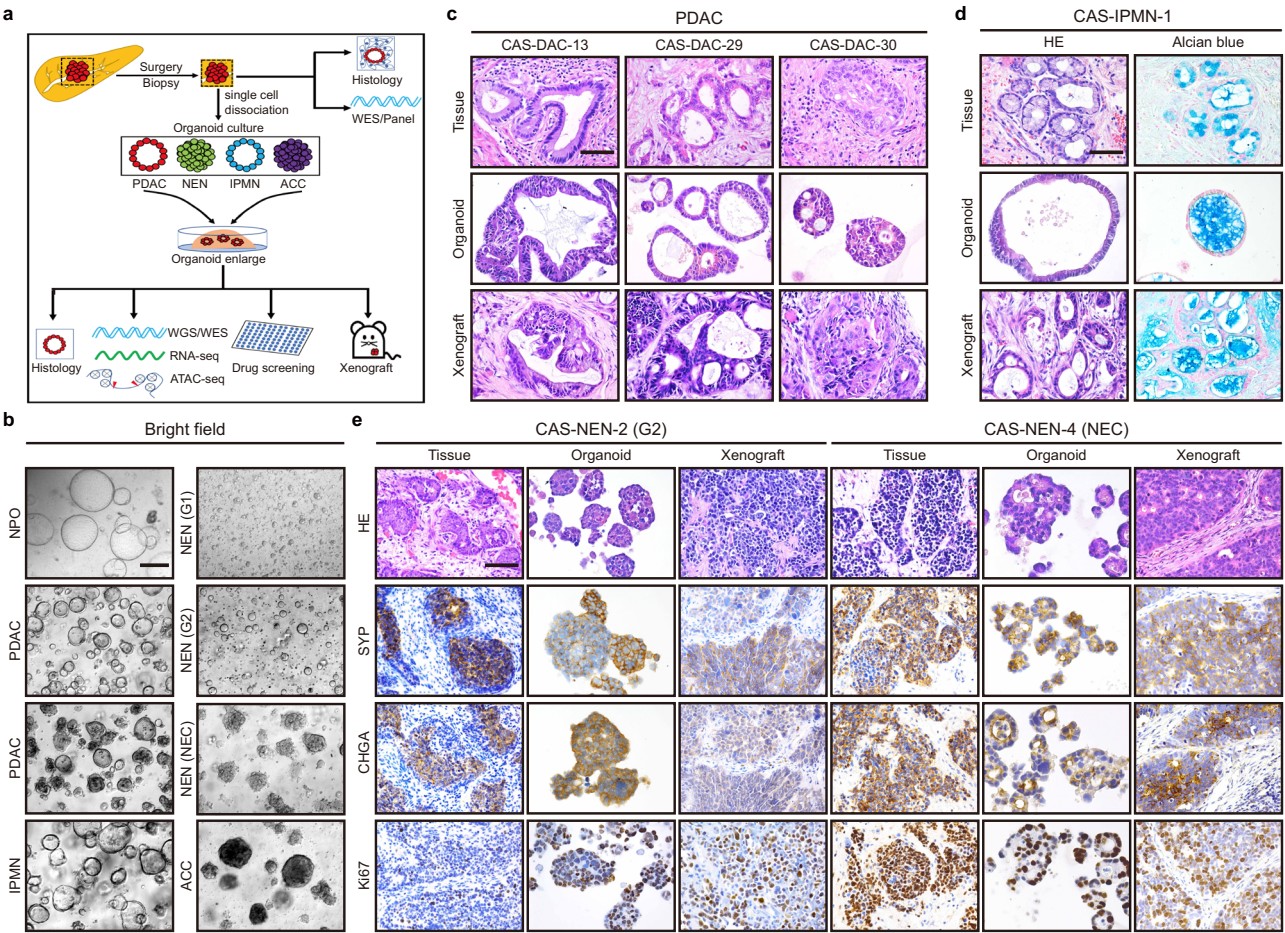

**Fig. 1 Establishment and histological characterization of PDPCOs. a** Overview of the study. A total of 84 PDPCO lines, including PDAC, NEN, IPMN and ACC, were established by samples obtained by surgery or EUS-FNA. Original tissues were used for histological examination and WES/panel sequencing. PDPCOs were subjected to engraftment, histological examination, WGS/WES, RNA-seq, ATAC-seq and drug screening. **b** Representative brightfield image of PDPCOs generated from normal pancreas, PDAC, IPMN, NEN and ACC tissue. Scale bar, 200 μm. Experiments are repeated at least three times with similar results. **c**, Representative H&E staining of PDAC primary tumors, organoids, and xenografts (CAS-DAC-13, CAS-DAC-29 and CAS-DAC-30). Scale bar, 50 μm. Experiments are repeated at least three times with similar results. **d** Representative H&E and alcian blue staining of an IPMN primary tissue, organoid and xenograft (CAS-IPMN-1). Scale bar, 50 μm. Experiments are repeated at least three times with similar results. **e** Representative H&E, SYP, CHGA and Ki67 staining of NEN primary tissues, organoids and xenografts (CAS-NEN-2 and CAS-NEN-4). CAS-NEN-2 was clinically diagnosed as WHO grade G2, while CAS-NEN-4 was diagnosed as NEC. Scale bar, 50 μm. Experiments are repeated at least three times with similar results. PDPCO, patient-derived pancreatic cancer organoid; PDAC, pancreatic ductal adenocarcinoma; NEN, pancreatic neuroendocrine neoplasm; IPMN, intraductal papillary mucinous neoplasm; ACC, acinar cell carcinoma; EUS-FNA, endoscopic ultrasound-guided fine needle aspiration biopsy.

an endogenous mutational process initiated by spontaneous deamination of 5-methylcytosine. Based on signature exposure, the 70 exocrine organoids were divided into three clusters, among which the cluster enriched for signature C accounted for the majority of exocrine organoids (47/70) and signature B exhibited notable advantages in only two organoid lines (2/70) (Supplementary Fig. 2d). These results revealed the commonly enriched gene mutational signatures in exocrine organoids and Cosmic signature 1, as well as Cosmic signature 5 achieved the major compositions.

Among the 70 organoid lines, enough fresh frozen tumor tissue was available to perform whole-exome sequencing (WES) on 10 specimens. We firstly examined the purity of PDPCOs and tissues, and further compared their concordance in mutations among exon regions. As expected, the estimated cellularity of theses tissues was relatively high at around 40%, since we have selected samples with enough tumor cells based on histological examination (Supplementary Fig. 2e). Besides, extremely high purity was observed in the PDPCOs, which has been reported one of the advantages for the organoid model (Supplementary Fig. 2e).

Comparison of detailed mutations among exon regions between organoids and matched tissues showed that the majority of the 10 pairs achieved favorable concordance of 50–60%, and the organoid line derived from ACC which was characterized by abundant tumor cell exhibited the highest concordance of 74.60% (Supplementary Fig. 2e). Moreover, we found that most of the mutations in exon regions of tumor tissues could be similarly detected in their corresponding organoid lines, in which 6 pairs reached to more than 70% (Supplementary Fig. 2e, f). Then we compared the mutational landscape between the original tumors and the patient-derived organoids and found that the distribution of variation within the coding region was similar between the PDPCOs and corresponding tissue samples (Supplementary Fig. 2g). In addition, the six top mutated genes, *KRAS*, *TP53*, *SMAD4*, *CDKN2A*, *RNF43* and *ARID1A*, were examined for their consistency between the PDPCOs and corresponding tissue samples (Supplementary Fig. 2h). The vast majority of these mutations were also observed in the matched tissue samples, except for one *SMAD4* mutation in CAS-DAC-16, which might

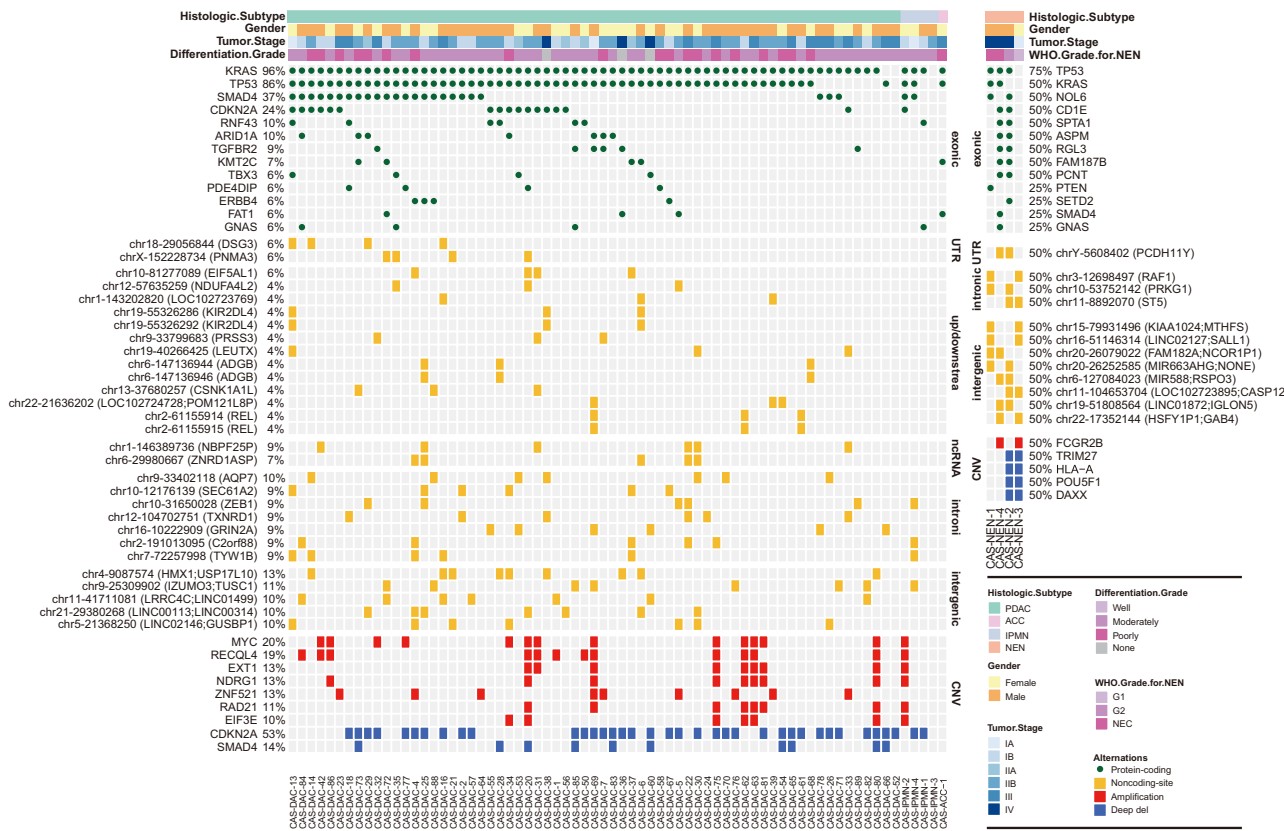

**Fig. 2 WGS analysis of genomic mutations and alterations in PDPCOs.** Genomic landscape of 70 exocrine PDPCOs, including PDACs, IPMNs, and ACCs (left), and 4 neuroendocrine PDPCOs (right). The figure shows exonic mutations; noncoding mutations in UTR, upstream/downstream, intronic, ncRNA and intergenic regions; and CNVs. CAS-DAC-23 with WES of blood DNA; CAS-NEN-2 without sequencing of blood DNA. PDPCO patient-derived pancreatic cancer organoid, PDAC pancreatic ductal adenocarcinoma, NEN pancreatic neuroendocrine neoplasm, IPMN intraductal papillary mucinous neoplasm, ACC acinar cell carcinoma, CNV copy number variation.

be due to tissue heterogeneity or the higher tumor purity of the organoid model or could be an additional mutation acquired during culture. Moreover, somatic CNV patterns were similar between the organoid lines and original patient samples (Supplementary Fig. 2i).

The 4 neuroendocrine tumor-derived organoid lines were also subjected to WGS; among these lines, CAS-NEN-2 lacked germline information (Fig. 2). Two organoid lines, CAS-NEN-1 and CAS-NEN-4, generated from NECs, carried mutations in both *KRAS* and *TP53*, which are common in malignant NENs[26,27]. In addition, CAS-NEN-1 also had a mutation in *PTEN*, while CAS-NEN-4 carried mutations in *SMAD4* and *GNAS*. However, these two NEC organoid lines showed limited variants in CNVs. In contrast, although the G1 organoid line CAS-NEN-3 exhibited very few mutational alteration, significant differences were observed in CNVs, including in *DAXX*, whose loss is closely correlated with pancreatic neuroendocrine tumorigenesis[28,29]. In addition to deep deletion of *DAXX*, the G2 organoid line CAS-NEN-2 also harbored a mutation in TP53, indicating aggressive biological behavior and consistent with its clinical metastasis.

Collectively, these results indicated that the PDPCO lines harbored copy number alterations highly representative of the overall landscape of pancreatic cancer as well as highly pancreatic cancer-specific gene mutations.

**PDPCOs reveal noncoding mutations in pancreatic cancer.** Both coding and noncoding driver mutations performed critical roles on tumorigenesis, while compared with coding tumor drivers which have been comprehensively investigated, the discovery

of noncoding driver mutations is particularly incomplete[30,31]. We further identified mutations in the noncoding region of 69 paired exocrine organoid-blood samples (Fig. 2 and Supplementary Data 2). A total of 527,272 noncoding mutations were identified in this cohort. Strikingly, many shared noncoding mutations were identified within noncoding RNA regions, regulatory regions (promoters, 5′ untranslated regions (UTRs), 3′ UTRs and enhancers), intergenic regions and intronic regions (Fig. 2). Most of these noncoding mutations have not been reported before, and their functions remain generally unknown. Further investigations to determine the roles of these noncoding mutations would certainly provide insights into pancreatic cancer.

**Transcriptomic profiling of PDPCOs identifies four types of exocrine pancreatic cancer.** PDAC is a deadly malignancy that has been classified into two major subtypes: basal-like and classical[3,4]. RNA-seq was performed on 84 PDPCOs and 3 NPOs, among which 45 pancreatic cancer organoid lines (39 CAS-DACs, 1 CAS-IPMN, 1 CAS-ACC and 4 CAS-NENs), as well as 3 NPOs were first used as a discovery cohort to explore transcriptomic subtypes. As expected, the NPOs were clearly separated from the pancreatic cancer organoids, while the pancreatic cancer organoids were divided into an exocrine group and a neuroendocrine group by principal component analysis (PCA) (Fig. 3a). Representative genes of acinar cell, neuroendocrine cell and PDAC showed significant higher expression in matched organoids (Supplementary Fig. 3a)

Using NMF clustering, we identified four PDAC subtypes in the exocrine group, not including the CAS-ACC organoid line (Fig. 3b,

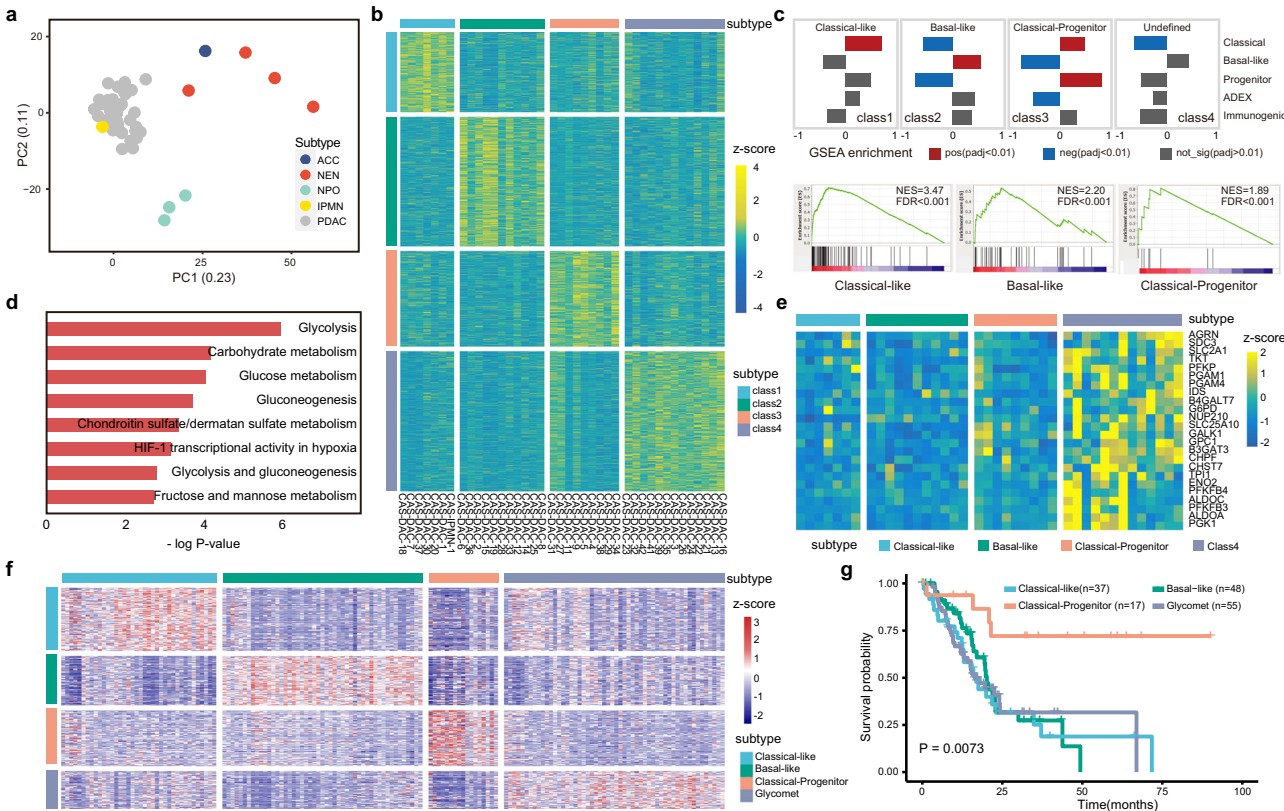

**Fig. 3 Transcriptome analysis of PDPCOs. a** PCA of gene expression for 45 PDPCOs in the discovery cohort: 39 CAS-DACs, 4 CAS-NENs, 1 CAS-IPMN and 1 CAS-ACC. **b** Heatmap of the four transcriptomic subtypes in the discovery cohort of 40 exocrine PDPCOs (39 CAS-DACs and 1 CAS-IPMN), as revealed by unsupervised NMF clustering. Samples are listed in columns, and signature genes are listed in rows as z-scores. **c** Identification of biological characteristics in each subtype by previously reported signatures. A *P*-value < 0.01 was used to determine positive or negative enrichment of published signatures in a subtype (top). Detailed GSEA results are shown at the bottom. **d** Bar graph showing the results of enrichment analysis of class 4 DEGs by Enrichr. Significance was computed by one-sided hypergeometric test. **e** Heatmap indicating the expression of glycolysis pathway-related genes across the four subtypes. Samples are listed in rows, and signature genes are listed in columns as z-scores. **f** External validation of the transcriptomic subtypes in the TCGA PAAD cohort. Samples are listed in rows, and signature genes are listed in columns as z-scores. **g** Kaplan-Meier survival analysis showing the overall survival characteristics of patients with the four transcriptomic subtypes, as identified in **f**. *P* value is determined by using log-rank test. PDPCO, patient-derived pancreatic cancer organoid; PDAC, pancreatic ductal adenocarcinoma; NEN, pancreatic neuroendocrine neoplasm; IPMN, intraductal papillary mucinous neoplasm; ACC, acinar cell carcinoma; NPO, normal pancreatic ductal organoid; PCA, principal component analysis; NMF, nonnegative matrix factorization; GSEA, gene set enrichment analysis; DEG, differentially expressed gene.

Supplementary Fig. 3b and Supplementary Data 3). Based on differentially expressed genes (DEGs) and gene set enrichment analysis (GSEA) with 5 previously identified subtypes[3,4,8], we revealed the biological characteristics of the four subtypes. Class 1 and class 2 were named the classical-like and basal-like types of PDAC and were predominantly composed of classical signatures and basal-like signatures, respectively (Fig. 3c). Class 3 was significantly enriched with both classical- and progenitor-type signatures and named the classical-progenitor type of PDAC (Fig. 3c). Class 4 constituted a new type of PDAC that did not match any of the reported subtypes. GSEA revealed that class 4 was highly associated with glycometabolic pathways, such as glycolysis, carbohydrate metabolism and glucose metabolism (Fig. 3d). Consistent with this finding, glycolysis pathway-related genes were highly expressed in class 4 (Fig. 3e). Therefore, we named class 4 the glycomet type of PDAC.

To verify the classification of the PDAC subtypes, the remaining 39 organoids (36 CAS-DACs and 3 CAS-IPMNs) were used for internal validation. The classical-like, basal-like and glycomet types were successfully identified (Supplementary Fig. 3c and Supplementary Data 3). However, the classical-progenitor type was not reproduced, possibly because of the limited size of this cohort. Furthermore, the four subtypes of PDACs were

independently confirmed in the recently published TCGA PAAD dataset[23] (Fig. 3f).

We next examined the clinical significance of these PDAC transcriptomic subtypes. Previously, the classical subtype was found to have a markedly better outcome than the basal-like subtype[3,4,32] (Supplementary Fig. 3d). Strikingly, in our study, only the classical-progenitor type showed a prognosis significantly better than that of the other subtypes, while classical-like, basal-like and glycomet types had similar poor survival outcomes (Fig. 3g and Supplementary Fig. 3e). These results demonstrated that a glycomet type of PDAC revealed in our PDPCOs has a high level of glycometabolic activity and the classical-progenitor type of PDAC has a better prognosis.

**The chromatin accessibility landscape provides new insights into the gene regulatory network of pancreatic cancer.** We profiled the chromatin accessibility landscape of 41 exocrine PDPCOs (39 CAS-DACs, 1 CAS-IPMN and 1 CAS-ACC) and 4 NEN organoid lines (CAS-NEN-1, CAS-NEN-2, CAS-NEN-3, CAS-NEN-4) by overcoming a daunting task due to the intense desmoplastic stroma in pancreatic cancer with ATAC-seq. A total of 156,721 and 26,975 reproducible (observed in more than one organoid) chromatin accessibility peaks were identified in the 41 exocrine pancreatic tumor organoids and 4 NEN organoid lines,

respectively. We merged these chromatin accessibility peaks (162,026) for comparison with previous pan-cancer data[12] (Supplementary Fig. 4a), and we observed similar enrichment in transcription start site (TSS) and enhancer (Enh) regions between the PDPCO and pan-cancer data (Supplementary Fig. 4b). As expected, in principal coordinates analysis, the PDPCOs were markedly separated from the other cancers (Supplementary Fig. 4c).

To comprehensively understand the underlying cell types of pancreatic tumors, we integrated the transcriptomic and chromatin accessibility data to explore the underlying key transcription factors (TFs). Based on the TF regulon set generated from the transcriptomic data, we identified cluster-specific TF regulons by motif enrichment according to the chromatin accessibility data (Fig. 4a and Supplementary Data 4). Comparison between exocrine and neuroendocrine tumor organoid lines showed significant enrichment of neuroendocrine cell lineage-associated TFs, such as ASCL1, NEUROD1, NKX2-5 and POU3F2, in neuroendocrine organoids (Fig. 4b)[20]. In addition, neuroendocrine tumor organoids were markedly enriched with the NKX6-1 regulon (Fig. 4b). Previous studies have demonstrated that NKX6-1 is a homeobox TF participating in the development and regulation of the endocrine function of pancreatic islets[33], which further helps to reveal the biological characteristics of pancreatic neuroendocrine tumor organoids.

We next investigated the heterogeneity of TF regulons among the four subtypes of PDAC, where the enriched TFs could regulate as many as 40%~50% of the DEGs between subtypes (Fig. 4c, Supplementary Fig. 4d and Supplementary Data 4). Specifically, in the classical-like subtype, we found significant enrichment of HNF4A, which was demonstrated to be a hallmark of the classical subtype of pancreatic cancer[10]. Additionally, enrichment of HNF4G was found in this subtype. Previous studies have suggested that overexpression of HNF4G can promote PDAC invasion[34]. TP63, a well-defined basal/squamous cell marker[3], showed higher enrichment in the basal-like subtype. Consistent with the poor prognosis of the basal-like subtype, we revealed notable enrichment of FOXA1, which was showed to promote the metastasis of pancreatic cancer[11]. Surprisingly, the HOX gene family was also significantly enriched in the basal-like subtype. Many HOX genes, such as HOXA13, HOXB7 and HOXB8, were previously reported to be upregulated in pancreatic cancers with increased proliferative and metastatic abilities[35], and we first revealed their common enrichment in the basal-like subtype of pancreatic cancer. As expected, in the classical-progenitor subtype, we observed enrichment of MYC, MYB and ATOH1, which have been previously identified as critical regulators of progenitor cells in the pancreas, colonic crypt and intestinal epithelium[36–38], reinforcing the specific progenitor feature of this subtype. In the newly discovered glycomet subtype, we found many enriched TFs such as NFE2, MAFK and PAX8, which have been demonstrated to played critical roles in other aggressive solid cancers, facilitating a deeper understanding for this subtype. Recent studies proposed that overexpression of NFE2 could significantly enhance the metastasis capability of triple-negative breast cancer by mimicking the bone microenvironment and activating Wnt pathway[39]. Similarly, MAFK was also reported to promote the progression of triple-negative breast cancer, in which MAFK notably induced epithelial-mesenchymal transition (EMT) and tumor invasion by regulating the targeted gene GPNMB[40]. In addition, a potential oncogenic TF in renal and ovarian cancers-PAX8[41] appeared in the glycomet subtype, indicating its important contribution to the malignant behavior of this class. These results provide new comprehensive insights into the biological heterogeneity of pancreatic cancer from both the transcriptomic and chromatin accessibility perspectives and reveal subtype-specific TFs.

To more completely decipher noncoding genome in exocrine pancreatic tumors by associating ATAC-seq peaks and the genes, we correlated the chromatin accessibility peak and gene expression data across the 41 exocrine pancreatic tumor organoids using the published strategy[12,42]. We identified 2,257 ATAC-seq peak-to-gene links, which could be divided into three types: (i) 154 promoter peak-to-gene links, (ii) 1,462 positive distal ATAC-seq peak-to-gene links, and (iii) 641 negative distal ATAC-seq peak-to-gene links (Fig. 4d and Supplementary Data 5). As expected, we found a notable difference between promoter-to-gene links and distal ATAC-seq peak-to-gene links in the histogram showing the distance from a peak to its target gene. All links in promoter-to-gene correlation had no gaps, as expected, while links in the distal ATAC-seq peak-to-gene correlation decayed gradually with distance (Supplementary Fig. 4e, h). Further comparison revealed that most promoter-to-gene links were exclusively related (Supplementary Fig. 4f, g). In contrast, 95% of genes were regulated by no more than 6 different peaks in the distal ATAC-seq peak-to-gene links (Supplementary Fig. 4i), whereas 80% of peaks were predicted to link with a single gene (Supplementary Fig. 4j). This peak-to-gene linkage landscape not only provided new insights into the regulation of well-recognized driver genes such as CDKN2A (Supplementary Fig. 4k, l) but also predicted potential DNA regulatory elements of genes, such as GPANK1, which are largely unknown (Supplementary Fig. 4m). These results expanded our understanding of pancreatic cancer to include an analysis of these predicted peak-to-gene links and provided a comprehensive extension of the gene regulatory reference in pancreatic cancer and serves as resource to interpret noncoding somatic mutations.

**Integrated analysis reveals potential functional pancreatic cancer-related noncoding mutations**. Given the role of noncoding mutations as potential drivers of cancer initiation and progression[12,31], we performed integrated analysis of the WGS, ATAC-seq and RNA-seq data to identify regulatory noncoding mutations (Fig. 5a). The 31 exocrine pancreatic organoid samples (29 CAS-DACs, 1 CAS-IPMN, and 1 CAS-ACC) harbored a total of 240,900 called noncoding mutations. A total of 3.07% ($n = 7,398$) of these noncoding mutations were located within annotated ATAC-seq peaks (Fig. 5b and Supplementary Data 6), which potentially had functions in regulating chromatin accessibility and gene expression. In addition, 1,516 (20.49%) mutations correlated with a significant increase or decrease in chromatin accessibility at the mutation site compared with that of the other non-mutated organoids in this cohort (Fig. 5c).

Among these mutations, a mutation was identified within the enhancer of the RIMBP2 gene, which was reported to correlate with more favorable prognosis in lung squamous cell carcinoma patients[43]. This enhancer mutation was associated with a significant increase in chromatin accessibility at the mutated site and was accompanied by an increase in RIMBP2 mRNA expression (Fig. 5d, e). Notably, RIMBP2 expression was positively associated with pancreatic cancer patient survival in the TCGA PAAD cohort[23] (Fig. 5f), indicating a potential protective function of both this noncoding mutation and RIMBP2. In contrast, mutation of the S100A6 gene enhancer was associated with a decrease in chromatin accessibility at the mutation site and correlated with reduced S100A6 mRNA expression (Supplementary Fig. 5a, b). S100A6 functions in promoting cell invasion and EMT in pancreatic cancer[44,45], predicting worse patient outcomes with this mutation (Supplementary Fig. 5c). This mutation in the S100A6 gene enhancer could have functional consequences in pancreatic cancer. These

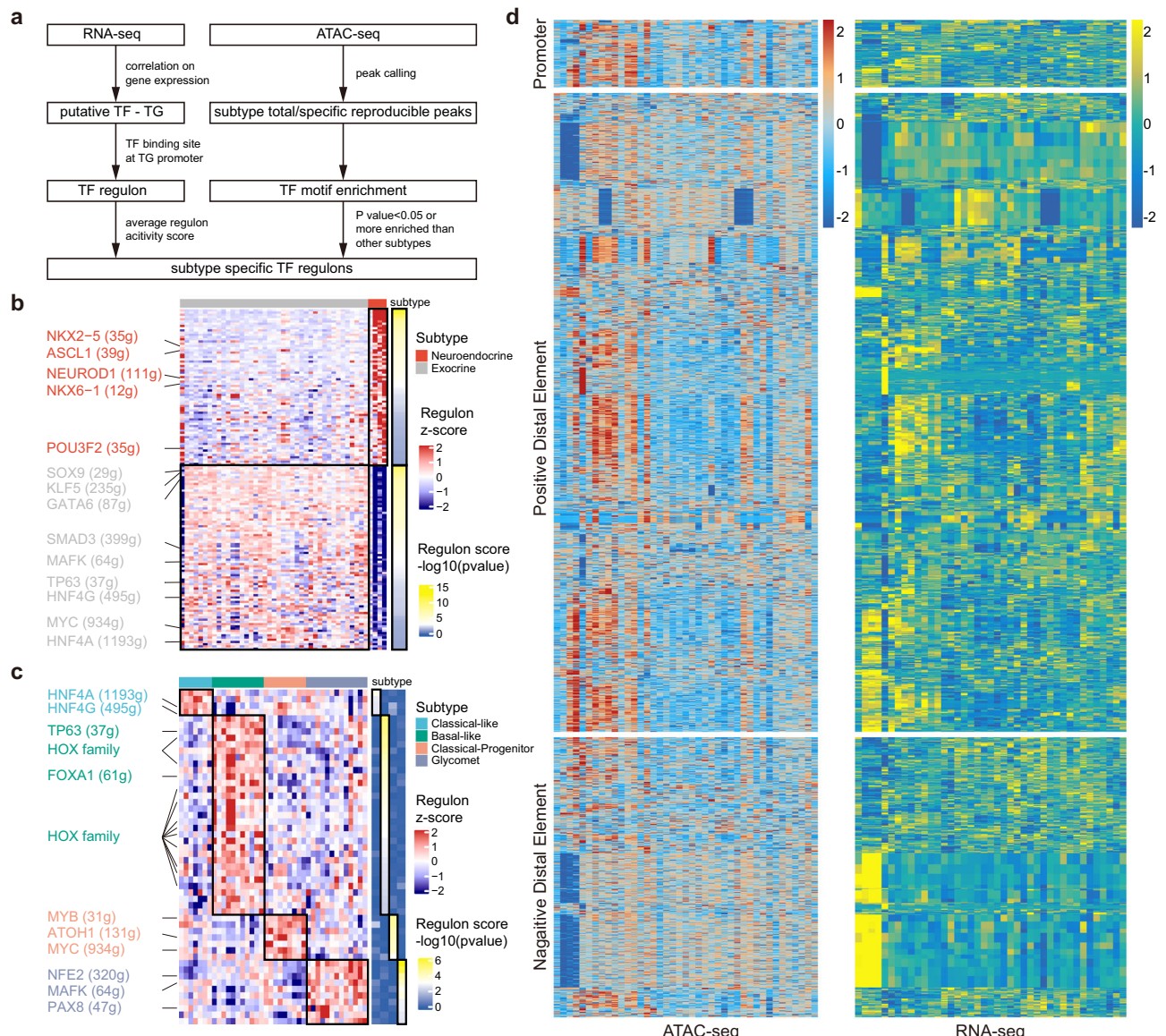

**Fig. 4 Integrated analysis of RNA-seq and ATAC-seq revealing TF regulons for PDPCO subtypes and putative peak-to-gene links. a** Schematic of the process for generating subtype-specific TF regulons in PDPCOs. Comparison of enriched TF regulons between exocrine and neuroendocrine PDPCOs in **b** and among the four PDAC transcriptome subtypes in **c**. The color of each heatmap on the left represents the regulon z-score. The color of each heatmap on the right represents the -log10(*P* value) of the regulon score. *P* values associated with one-sided unpaired *t* test were adjusted for multiple testing using FDR. Representative subtype-specific TF regulons are listed. **d** Heatmap of 2,257 putative links between ATAC-seq peaks (left) and genes (right) in 41 exocrine PDPCOs (39 PDACs, 1 IPMN, and 1 ACC). Each row represents an individual link between one ATAC-seq peak and one gene. The color of each heatmap represents the z-score for ATAC-seq peak's accessibility (left) or the z-score for gene expression (right). TF, transcription factor; PDPCO, patient-derived pancreatic cancer organoid; TG, target gene; PDAC, pancreatic ductal adenocarcinoma; FDR, false discovery rate.

results identified regulatory noncoding mutations that have potential as cancer drivers or diagnostic biomarkers.

**High-throughput drug screening reveals a chromatin accessibility signature associated with drug sensitivity**. To characterize the drug sensitivity signatures of pancreatic cancer, we performed primary drug screening on 35 exocrine PDPCOs (34 CAS-DACs and 1 CAS-IPMN) using a library of 283 chemicals targeting epigenetic-related signaling pathways, including those involved in histone modification, the cell cycle, DNA damage, some classical cellular signaling cascades, the cytoskeleton and other cellular aspects (Supplementary Fig. 5d and Supplementary Data 7). We first selected 87 chemicals that exhibited an inhibition rate of higher than 60% in at least one organoid. Among these 87

chemicals, 17 were excluded for similar inhibition rates among all screened organoids, and 11 were excluded for inadequate inhibition rates (the screening concentration was markedly higher than the reported IC50, while the inhibition rate was <80% in all screened organoids). The remaining 59 chemicals (Supplementary Fig. 5e, f and Supplementary Data 7) and 5 chemotherapeutic drugs used as first-line therapy for pancreatic cancer (gemcitabine, paclitaxel, 5-fluorouracil, oxaliplatin and irinotecan) were used for the secondary screening of 39 exocrine PDPCOs (38 CAS-DACs and 1 CAS-IPMN) (Supplementary Fig. 5f and Supplementary Data 8).

We next examined the association between ATAC-seq peaks and drug sensitivity. This analysis identified 15,397 links in this cohort, including both positive and negative correlations (Fig. 5g

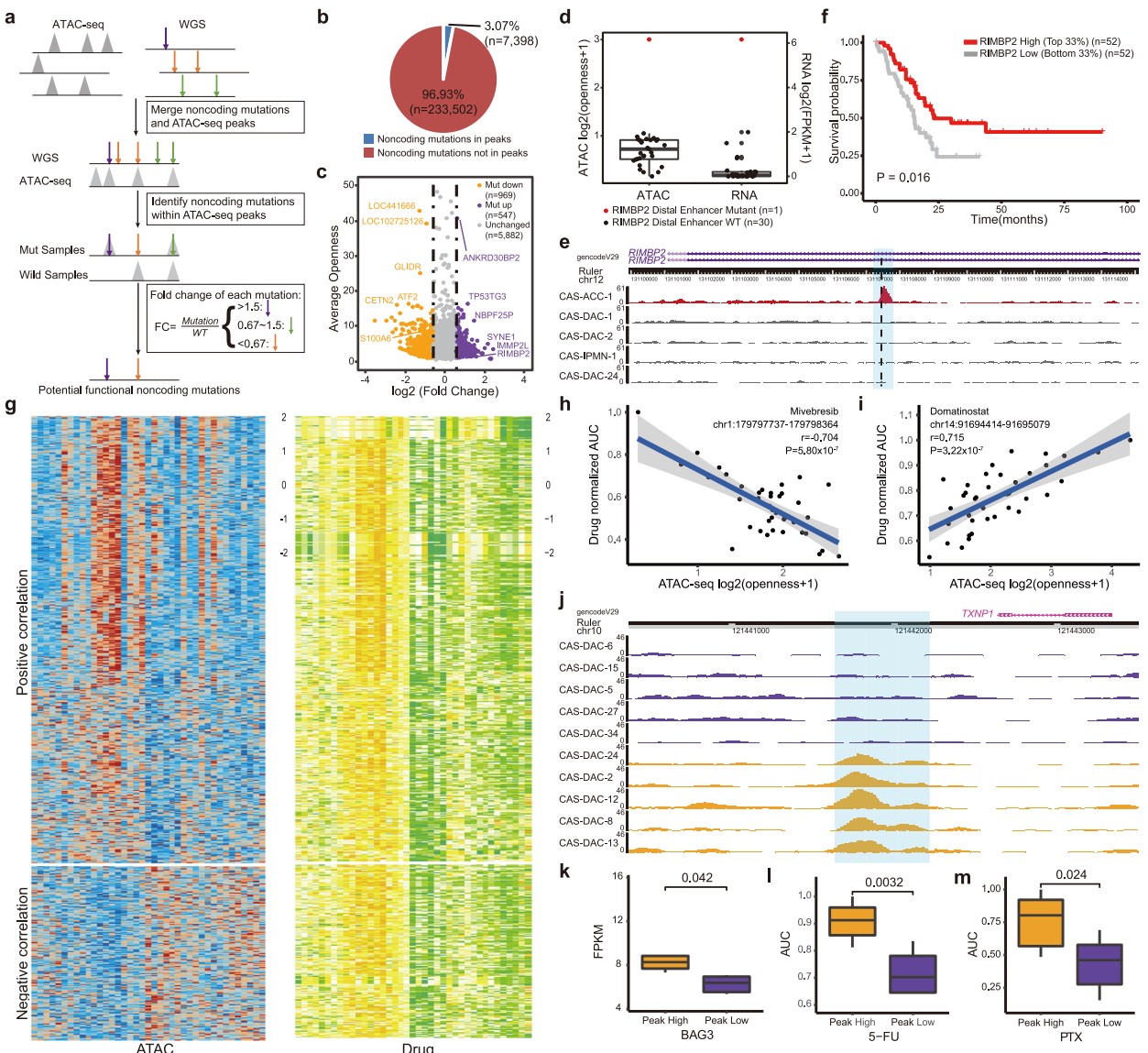

**Fig. 5 Integrated analysis of noncoding mutations and drug sensitivity in PDPCOs. a** Schematic for identifying potential functional noncoding mutations in 31 exocrine PDPCOs. **b** Proportions of noncoding mutations located within and outside ATAC peaks. **c** Dot plot showing fold change in accessibility at ATAC-seq peaks containing noncoding mutations across above 31 exocrine PDPCO. **d** Box plot showing chromatin accessibility at *RIMBP2* putative enhancer and *RIMBP2* gene expression across above 31 exocrine PDPCOs. Boxplot show the median (central line), the 25–75% interquartile range (box limits). **e** Normalized ATAC-seq tracks of *RIMBP2* putative enhancer locus in 5 representative samples. The red and gray tracks represent samples with and without mutation of the *RIMBP2* putative enhancer, respectively. Black dotted line indicates the position of the mutation, and the predicted enhancer region is highlighted by light blue shading. **f** Kaplan-Meier analysis of overall survival in the TCGA PAAD cohort stratified by high and low expression of *RIMBP2* gene. P value is determined by using log-rank test. **g** Heatmap of 15,397 putative peak-to-drug links in 39 exocrine PDPCOs. Each row represents an individual link between one ATAC-seq peak and one drug. The color represents the z-score for chromatin accessibility (left) or the z-score for drug AUC (right). Dot plot showing a representative negative correlation in **h** and a representative positive correlation in **i**. Significance is computed by Pearson's correlation coefficients without adjustment. **j** Normalized ATAC-seq tracks of *BAG3* putative enhancer locus in 10 representative samples. The yellow and purple tracks represent samples with and without peaks in *BAG3* putative enhancer, respectively. The peak region is highlighted by light blue shading. Comparison between peak high group with peaks in the *BAG3* putative enhancer and peak low group without such peaks in the above 10 samples of *BAG3* gene expression in **k**, AUC of 5-FU in **l** and AUC of PTX in **m**. Each group contains 5 biologically independent samples. Boxplot show the median (central line), the 25–75% interquartile range (box limits). Significance is computed by two-sided unpaired *t* test. PDPCO, patient-derived pancreatic cancer organoid; AUC, area under curve; 5-FU, 5-fluorouracil; PTX, paclitaxel.

and Supplementary Data 9). Strikingly, the chromatin accessibility peak at chr1:179797737-179798364 was negatively correlated with sensitivity to mivebresib, a bromodomain and extraterminal domain (BET) family inhibitor (Fig. 5h). In contrast, another chromatin accessibility peak at chr14:91694414-91695079 was positively correlated with sensitivity to domatinostat, a dual class I histone deacetylase (HDAC) and lysine demethylase inhibitor

(Fig. 5i). These results indicated the potential link between chromatin accessibility and drug sensitivity.

To understand the mechanisms underlying the abovementioned chromatin accessibility peak-to-drug links, we next investigated the link between these ATAC peaks and RNA expression. The ATAC peak at chr12:125033345-125034019 was positively correlated with the expression of the *NCOR2* gene (Ten

representative examples showing the most extreme differences in Supplementary Fig. 5g, h, general comparison in Supplementary Fig. 6a), which is involved in the regulation of the oncogenic JAK/STAT3 pathway in anaplastic large cell lymphomas and estrogen receptor-positive breast cancer[46,47]. As expected, this *NCOR2* gene-related ATAC peak predicted sensitivity to Go6976, an inhibitor of the PKC and JAK/STAT3 pathways (Ten representative examples showing the most extreme differences in Supplementary Fig. 5i, general comparison in Supplementary Fig. 6b). As another example, *MSH2* mRNA expression was significantly positively correlated with the ATAC peak at chr2:47355361-47355625 (Ten representative examples showing the most extreme differences in Supplementary Fig. 5j, k, general comparison in Supplementary Fig. 6c), which predicted sensitivity to HDAC inhibitors (panobinostat, abexinostat and quisinostat) (Ten representative examples showing the most extreme differences in Supplementary Fig. 5l-n, general comparison in Supplementary Fig. 6d). A previous study indicated that HDAC10 was the crucial enzyme for deacetylating MSH2 to promote DNA mismatch repair activity in cancer cells[48].

As cytotoxic neoadjuvant or adjuvant chemotherapy in combination with oncologic resection is still the most widely used treatment strategy for PDAC patients, we investigated the chromatin accessibility signature associated with sensitivity to cytotoxic chemotherapeutics. PDPCOs with higher chromatin accessibility at the peak at chr10:121441617-121442244 displayed significant resistance to 5-fluorouracil and paclitaxel (Ten representative examples showing the most extreme differences in Fig. 5j, l, m, general comparison in Supplementary Fig. 6f), and this peak positively correlated with *BAG3* gene expression (Ten representative examples showing the most extreme differences in Fig. 5k, general comparison in Supplementary Fig. 6e). Recent studies proved that BAG3 can effectively induce 5-fluorouracil and paclitaxel resistance in cancer cells[49,50]. Furthermore, we showed that the ATAC-seq peak at chr6:139728666-139729424 predicted sensitivity to oxaliplatin (Ten representative examples showing the most extreme differences in Supplementary Fig. 5o, q, general comparison in Supplementary Fig. 6h). This peak had a direct positive correlation with the expression of the *CITED2* gene (Ten representative examples showing the most extreme differences in Supplementary Fig. 5p, general comparison in Supplementary Fig. 6g), which was previously identified as a potential drug resistance gene for platinum[51]. These results indicated that chromatin accessibility peaks can be useful biomarkers for predicting chemosensitivity in pancreatic cancer and revealed the potential mechanism of drug sensitivity.

**PDPCOs recapitulate the patients' chemosensitivity.** Since chromatin accessibility could predict chemosensitivity in pancreatic cancer, we further investigated whether PDPCOs can accurately reflect the chemosensitivity profiles of pancreatic cancer patients. The results from high-throughput screening of chemotherapeutic drugs revealed significant interorganoid variability for a single agent and interagent variability within a single organoid, as similarly documented in clinical practice (Fig. 6a). In terms of chemosensitivity, we defined the organoids as sensitive (area under the curve (AUC) lowest 1st-13th), intermediate (AUC lowest 14th–26th) or resistant (AUC lowest 27th–39th) based on the dose-response curve and the corresponding AUC value (Fig. 6a). We explored the correlations in chemosensitivity between the organoids and patients. Clinical follow-up data of the 39 corresponding patients were collected from the prospective database of Changhai Hospital (Supplementary Data 10). Thirty-one patients underwent upfront radical surgery with subsequent adjuvant therapy, 6 patients received upfront radical surgery

without adjuvant therapy, and the other 2 patients received neoadjuvant therapy rather than upfront surgery (Fig. 6b). We divided the 31 patients into 3 groups based on the sensitivity of the paired organoids and the patients' actual adjuvant chemotherapy regimens: the sensitive group (patients treated with at least one chemotherapeutic agent to which the paired organoid was sensitive), the intermediate group (patients treated with at least one chemotherapeutic agent with an intermediate response in the paired organoid), and the resistant group (patients treated with chemotherapeutic agents to which the paired organoid was resistant) (Supplementary Fig. 7a). Survival analysis showed that the sensitive group had a significantly longer recurrence-free survival time than both the intermediate group (N/A vs 225 d, $P = 0.046$) and the resistant group (N/A vs 217 d, $P = 0.006$) (Fig. 6c). Comparison of imaging results between the sensitive group and resistant group at 6 months after surgery revealed that no evident recurrence occurred in patient CAS-DAC-24, who received gemcitabine; moreover, the paired organoid line (CAS-DAC-24) was sensitive to gemcitabine. In contrast, the organoid line CAS-DAC-22 was resistant to 5-fluorouracil, and the corresponding patient developed multiple liver metastases and local recurrence when 5-fluorouracil was used as adjuvant therapy (Fig. 6d). In the intermediate group, patient CAS-DAC-20 was treated with S1 (which has the same mode of action as 5-fluorouracil), to which the paired organoid CAS-DAC-20 had intermediate sensitivity, and exhibited liver metastasis upon imaging examination 6 months postsurgery (Fig. 6d). After the addition of another intermediate agent, gemcitabine, the lesion remained stable in terms of both tumor size and biomarker expression for approximately 4 months before a second progression (Supplementary Fig. 7b, c). Besides, we compared the image results of patients used the same chemotherapy agents but with different organoid sensitivity. S1 was used in CAS-DAC-5 (intermediate), CAS-DAC-20 (intermediate), CAS-DAC-21 (resistant), CAS-DAC-22 (resistant) and CAS-DAC-31 (sensitive), while GEM in CAS-DAC-24 (sensitive) and CAS-DAC-21 (resistant). Comparison of image examinations significantly demonstrated a strong consistency between organoid sensitivity and recurrence status under the same chemotherapy agent (Supplementary Fig. 7d, e), in which "sensitive" showed less recurrence than "intermediate", while "resistant" exhibited the most progression. These results indicated a high consistency in chemosensitivity between the organoids and paired patients.

We next evaluated the chemosensitivity of the PDPCOs in vivo using organoid-derived xenograft (ODX) models. CAS-DAC-18 (sensitive to gemcitabine), CAS-DAC-20 (intermediate sensitivity to gemcitabine and 5-fluorouracil), CAS-DAC-22 (resistant to 5-fluorouracil), CAS-DAC-24 (sensitive to gemcitabine) and CAS-DAC-36 (resistant to gemcitabine, 5-fluorouracil and paclitaxel) represented diverse groups of organoids with different chemosensitivities. As expected, ODX-18 (the CAS-DAC18-derived xenograft) and ODX-24 (the CAS-DAC24-derived xenograft) was highly sensitive to gemcitabine (Fig. 6e and Supplementary Fig. 8a-c), while ODX-20 (the CAS-DAC20-derived xenograft) was modestly sensitive to 5-fluorouracil and gemcitabine (Fig. 6f and Supplementary Fig. 8a), and ODX-22 exhibited notable resistance to 5-fluorouracil (Supplementary Fig. 8d, e). The reductions in tumor growth were 92.33% for ODX-18 treated with gemcitabine ($P < 0.001$), 64.50% for ODX-20 treated with gemcitabine ($P < 0.001$), 56.67% for ODX-20 treated with 5-fluorouracil ($P = 0.001$), 17.04% for ODX-22 treated with 5-fluorouracil ($P > 0.05$) and 75.58% for ODX-24 treated with gemcitabine ($P < 0.001$). In particular, ODX-36 displayed significant resistance to all of the three drugs (gemcitabine, 5-fluorouracil and paclitaxel; $P = 0.566$, $P = 0.183$ and $P = 0.077$), as demonstrated by the 7.79, 24.51, and 25.17% reductions in tumor growth in response to these respective drugs, which was consistent with the in vitro organoid drug screening results (Fig. 6g and Supplementary Fig. 8f).

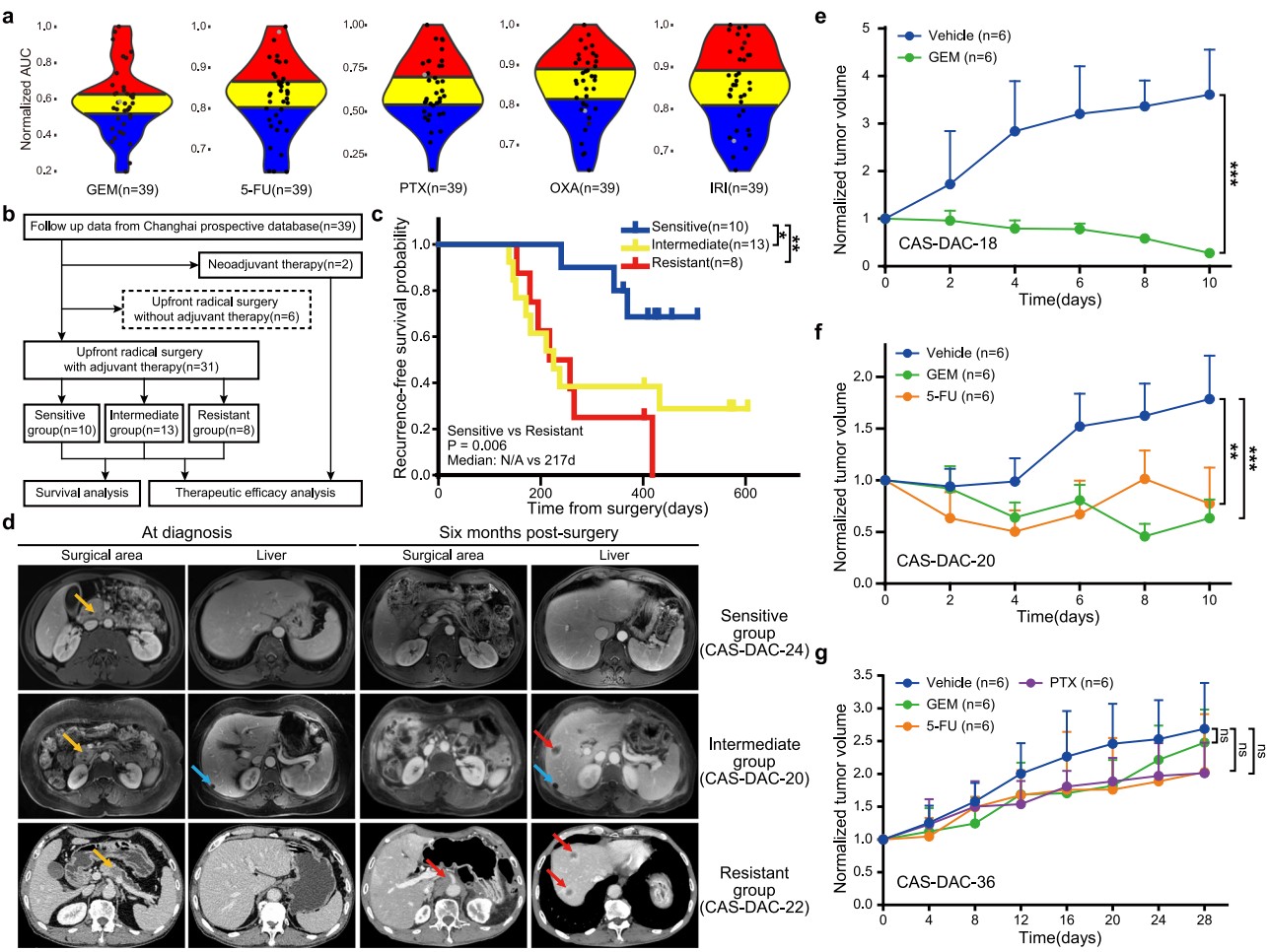

**Fig. 6 Clinically relevant chemosensitivity of PDPCOs with validation in in vivo xenograft models. a** AUCs of 5 chemotherapeutic agents in 39 exocrine PDPCOs (38 CAS-DACs and 1 CAS-IPMN). With respect to each chemotherapeutic agent, the 39 PDPCOs were divided equally into sensitive (blue, AUC lowest 1st-13th), intermediate (yellow, AUC lowest 14th-26th) and resistant (red, AUC lowest 27th-39th) groups. The gray dot indicated CAS-DAC-14. **b** Diagram of the clinical analysis workflow. **c** Kaplan-Meier survival analysis showing the recurrence-free survival outcomes of the patients corresponding to the 39 PDPCOs. $P$ value is determined by using log-rank test. Based on the consistency between the clinical adjuvant chemotherapy regimen and chemosensitivity of the matched organoid, three groups were identified: sensitive ($n = 10$), intermediate ($n = 13$) and resistant ($n = 8$). **d** Representative radiation examination of both the surgical area and liver in the sensitive group (CAS-DAC-24, CE-CT), intermediate group (CAS-DAC-20, CE-MRI), and resistant group (CAS-DAC-22, CE-MRI) at the time of diagnosis and six months post-surgery. Yellow arrow, primary tumor; Blue arrow, hepatic cyst; Red arrow, metastasis. **e** Drug test of ODX-18 with GEM (n=6), using Vehicle as a control (n=6). Data are presented as mean values + SEM. Statistical Significance was computed by two-sided unpaired $t$ test. Source data are provided as a Source Data file. **f** Drug test of ODX-20 with GEM ($n = 6$) and 5-FU ($n = 6$), using Vehicle as a control ($n = 6$). Data are presented as mean values + SEM. Statistical Significance was computed by two-sided unpaired $t$ test. Source data are provided as a Source Data file. **g** Drug test of ODX-36 with GEM ($n = 6$), 5-FU ($n = 6$) and PTX ($n = 6$), using Vehicle as a control (n=6). Data are presented as mean values + SEM. Statistical Significance was computed by two-sided unpaired $t$ test. Source data are provided as a Source Data file. Statistical analysis, ns $P \geq 0.05$, *$P < 0.05$, **$P < 0.01$, ***$P < 0.001$. PDPCO, patient-derived pancreatic cancer organoid; AUC, area under curve; GEM, gemcitabine; 5-FU, 5-fluorouracil; PTX, paclitaxel; OXA, oxaliplatin; IRI, irinotecan; CE-CT: contrast-enhanced computed tomography; CE-MRI: contrast-enhanced magnetic resonance imaging.

These results demonstrated that PDPCOs can reliably recapitulate patient-specific and clinically relevant responses to chemotherapy.

Moreover, we tried to expand these observations to the neoadjuvant therapy setting. Organoid CAS-DAC-38 was generated by EUS-FNA biopsy before treatment and showed sensitivity to gemcitabine and paclitaxel and resistance to 5-fluorouracil in chemotherapeutic drug screening (Supplementary Fig. 9a). During neoadjuvant therapy in patient CAS-DAC-38, although the level of the pancreatic cancer cell marker cancer antigen (CA) 19-9 was persistently above the upper limit of detection (>1200 U/mL), the levels of carcinoembryonic antigen (CEA) and CA 72-4 notably declined during combination chemotherapy with gemcitabine and paclitaxel (Supplementary Fig. 9b). However, tumor growth was no longer suppressed after the patient switched to combination chemotherapy with 5-fluorouracil and paclitaxel due to the intolerable side effects of combination chemotherapy with gemcitabine and paclitaxel (Supplementary Fig. 9b). The chemosensitivity profiles of patient CAS-DAC-38 and the paired organoid CAS-DAC-38 showed high consistency in the clinical setting of neoadjuvant therapy.

## Discussion

Epigenetic characteristics have shown that chromatin states, such as chromatin accessibility, histone modifications and chromatin interactions, play a critical role in dissecting cellular heterogeneity[52]. For example, the accessible pan-cancer genome provides comprehensive information on the prognosis and

potential therapeutic strategies of diverse cancer types[12]. Primary PDAC tissue contains an excessive population of stromal cells, which has been an obstacle in characterizing chromatin accessibility. Here, we generated a pancreatic cancer organoid biobank and completed an initial characterization of the chromatin regulatory landscape of human pancreatic cancer. Integration of the chromatin accessibility landscape with WGS, transcriptome and high-throughput drug sensitivity profiling data identified gene regulatory network underlying pancreatic cancer, expanded the repertoire of functional noncoding mutations in pancreatic cancer, and revealed the potential ability of chromatin accessibility to predict drug sensitivity networks. These data may constitute a fundamental research resource that facilitates the discovery of potential treatment targets and drugs for pancreatic cancer.

The discovery of driver mutations in both protein-coding and noncoding regions has critical implications for patients with cancer[30,31]. Previously, mutations in the coding regions of the pancreatic cancer genome have been investigated using patient-derived organoid models[8,17,19]. In this study, we found that driver mutations in protein-coding regions were consistent with those in primary tumors and a published database[23]. Furthermore, we investigated the mutations in noncoding regions and identified 527,272 noncoding mutations. We integrated WGS, ATAC-seq and RNA-seq data to identify regulatory noncoding mutations with corresponding chromatin accessibility and expression changes[53]. Strikingly, more than 3% of the noncoding mutations were located within chromatin accessibility peaks, and more than 20% (1527 of 7470 cases) of these noncoding mutations significantly changed chromatin accessibility at the peaks in which they were included, compared with that in the non-mutated genes in this cohort. For example, noncoding mutations in the enhancer region of RIMBP2 were highly associated with elevated chromatin accessibility at the mutation site, which promoted the expression of RIMBP2, leading to improved patient prognosis. Notably, noncoding mutations without significant changes in chromatin accessibility at their peaks, especially highly recurrent noncoding mutations, might exert their influence through mechanisms other than affecting chromatin accessibility[54]. In general, the integrated multi-omics data in the biobank provided a systematic approach for investigating the regulatory function across the noncoding genome and could effectively facilitate the understanding of the roles of noncoding mutations in promoting and suppressing tumor development.

The transcriptomic classification of pancreatic cancer has been intensively investigated. Via microdissection to enrich epithelial cells, pancreatic cancer can be divided mainly into the classical and basal-like subtypes[2], which can also be observed in several pancreatic organoid biobanks[8,17]. Recently, genomic aberrations and metabolic classifications have also provided novel biological insight into previously established PDAC subtypes[7,32]. In this study, we identified four transcriptomic subtypes incorporating three known subtypes (classical-like, basal-like and classical-progenitor) and one new subtype, glycomet, which is highly associated with glycometabolic pathways. Survival analysis demonstrated that the classical-progenitor subtype significantly correlated with better outcomes, unlike the other subtypes, indicating the prognostic value of this classification. Furthermore, we revealed the potential underlying regulatory mechanism by integrated analysis of TF regulons and the binding motifs of chromatin accessibility peaks. These results provided insights into the transcriptomic characteristics of pancreatic cancer and further revealed the underlying regulatory mechanism of different subtypes. The integrated profiling confirmed TFs, which could facilitate a comprehensive understanding of the distinct biological behavior of pancreatic cancer and potentially promote the clinical utility of transcriptomic subtypes.

Epigenetic aberrations are potential therapeutic targets that could be pharmacologically reversible, whereas genetic changes are commonly difficult to reverse[55]. In this study, we performed high-throughput drug screening on pancreatic cancer organoids using a library of 283 chemicals targeting epigenetic-related signaling pathways. The chromatin accessibility peak-to-gene linkage data demonstrated that these peak-to-gene links are new biomarkers for predicting sensitivity to epigenetic-related chemicals. In addition, similar ATAC peak-to-gene biomarkers for the chemosensitivity of pancreatic cancers were observed in this study. Based on this high-throughput drug screening and integrated profiling of PDPCOs, we identified that ATAC-seq peaks could be used to predict drug sensitivity, which may promote the exploration of new treatment strategies and the development of personalized therapies.

Previously, transcriptomic signatures for the five chemotherapeutic agents were generated to predict the specific sensitivity of pancreatic cancer patients using organoid cultures[8]. However, sufficient evidence proving the consistency between the pancreatic cancer organoids and the clinical responses of the corresponding patients was noticeably lacking. We compared organoid sensitivity with the clinical responses of the paired patients in a large cohort and confirmed significant survival benefits in patients who were treated with chemotherapeutics to which the corresponding organoid was sensitive. Furthermore, we validated and quantified organoid chemosensitivity in vivo with ODX models. We confirmed the strong correlation between organoid chemosensitivity and clinically relevant responses, which formed a solid foundation for guiding individualized chemotherapy in pancreatic cancer with PDPCOs in drug screening and clinical practice.

Pancreatic NENs are heterogeneous tumors with a highly variable prognosis. Patients with lower grade (G1/G2) tumors were reported to have more favorable survival outcomes than those with higher grade (G3/NEC) tumors[56], while a small subset of G1/G2 NENs were also aggressive and had a chance of metastasis[57]. Recently, research on pancreatic NENs has attracted much attention but has been restricted by the limited availability of models[20,58]. In this study, we generated 4 pancreatic NEN organoids with the capability of continuous passage in vitro. By histological and molecular analysis, we confirmed the NEN features of these 4 organoid lines. In particular, although DAC-47 was pathologically diagnosed as G2, it harbored a TP53 mutation, which is a common mutation in G3/NEC tumors[20,27], indicating aggressive biological behavior consistent with its clinical metastasis. DAC-46 was also generated from a metastasis, but unlike DAC-47, it harbored mutations in KRAS, TP53, and PTEN, which is another typical characteristic of malignant NENs[26]. These 4 organoid lines are critical supplements for the research model of pancreatic NEN, which could help us to better comprehend the aggressive biological behavior of NEN and facilitate the development of clinical treatment.

In conclusion, integrated profiling of the PDPCO biobank identified the potential regulatory function of noncoding mutations and the gene regulatory network underlying pancreatic cancer and revealed peak-to-gene links as new biomarkers for predicting drug sensitivity. The clinical responses of the paired patients further confirmed the advantages of organoid models in predicting therapeutic outcomes. Our study revealed a systematic approach to comprehensively understand the biology of pancreatic cancer and extended the utility of pancreatic cancer organoids.

## Methods

**Samples and patients**. Tumor samples were collected from pancreatic cancer patients who received surgical resection or EUS-FNA at Changhai Hospital. The

parts distant from the tumor in surgical specimens were used as normal samples. Samples were pathologically confirmed as tumor or normal tissue. Clinical variables of patients were collected from the prospective database of Changhai Hospital. All patients provided written informed consent for the use of their clinical data and surgical specimens, and consent to publish clinical information potentially identifying individuals was obtained. The study was conducted in accordance with the national guidelines and was approved by the Ethics Committee of Changhai Hospital (Approval number: CHEC2018-111). In addition to approval by the local IRB, this study has been reviewed by and is compliant with the Chinese Ministry of Science and Technology (MOST) for the Review and Approval of Human Genetic Resources (approval no. 2021BAT1264).

**Organoid culture**. The following culture media were used: basic medium (advanced DMEM/F12, 10 mM HEPES, 1X GlutaMAX-I, 100 µg/ml Primocin, 1X penicillin/streptomycin solution) and complete medium (advanced DMEM/F12, 10 mM HEPES, 1X GlutaMAX-I, 100 µg/ml Primocin, 1X penicillin/streptomycin solution, 500 nM A83-01, 10 µM Y-27632, 1.56 mM N-acetylcysteine, 10 mM nicotinamide, 10 ng/ml FGF10, 1X B27 supplement, 10 µM forskolin, 30% Wnt3A conditioned medium, 2% R-spondin conditioned medium, 4% Noggin conditioned medium). For surgical samples, tissues were minced and incubated in digestion medium (2.5 mg/ml collagenase II and 10 µM Y-27632 in basic medium) at 37 °C with mild agitation for approximately 1 h. The obtained cells were cultured on suspension plates with Matrigel and complete medium (PDAC samples were cultured in complete medium, while samples of normal pancreas and other subtypes of pancreatic cancer were cultured in complete medium supplemented with EGF 50 ng/ml). Material obtained from biopsy samples was directly cultured under the above conditions without digestion. The media used for organoid cryopreservation were composed of the corresponding culture medium (90%) and 10% DMSO. Detailed information of involved reagents was provided in Supplementary Data 11. The established organoids were routinely tested for mycoplasma contamination. All organoid experiments were performed at the Shanghai Institute of Biochemistry and Cell Biology.

**Histology and immunohistochemistry**. Tissues and xenografts were fixed with 4% paraformaldehyde overnight at 4 °C, while PDPCOs were fixed with 4% paraformaldehyde for 15 min at room temperature. The samples were then embedded in paraffin before slicing to a 5-µm thickness. All sections were dried at 60 °C for 120 min and incubated in xylene before dehydration in alcohol. The sections were either stained with H&E or processed for immunohistochemistry. After incubation in 0.1 mol/L citrate buffer (pH 6.0) in a boiling water bath for 20 min, endogenous peroxidase activity was blocked with 3% hydrogen peroxide for 15 min, and nonspecific binding was further blocked with 5% goat serum for 60 min. The sections were incubated with the primary antibody for 120 min at room temperature and were then washed and incubated with the secondary antibody for 60 min at 37 °C. Detection was performed with DAB prior to hematoxylin counterstaining, dehydration, clearing and mounting. The antibodies used for staining organoids, tissues and xenografts were as follows: anti-CK19 (1:500, D4G2, CST), anti-Ki67 (1:500, SP6, Abcam), anti-α1-ACT (1:150, ZA-0006, ZSGB-BIO), anti-Bcl-10 (1:50, ab33905, Abcam), anti-CHGA (1:3000, SP-1, ImmunoStar), and anti-SYP (1:250, RM-9111-S, Thermo).

**Whole-genome library preparation and sequencing**. Genomic DNA from PDPCOs and blood was extracted using a QIAamp DNA kit (Qiagen, 51306). Whole-genome libraries were generated using a NEBNext Ultra DNA Library Prep Kit (New England Biolabs, E7370L) according to the manufacturer's protocols. A 1 µg aliquot of DNA was used as input for fragmentation into ~300 bp fragments using a Covaris LE220 sonicator, and purification was performed with DNA Clean Beads. The DNA fragments underwent bead-based size selection and subsequent end repair, adenylation, and ligation to Illumina sequencing adapters. The final libraries were evaluated using a QIAxcel bioanalyzer. Libraries were sequenced on the Illumina HiSeq XTen platform aiming for 30X and 50X coverage for blood and organoids, respectively, and 150 bp paired-end reads were generated.

**Whole-exome library preparation and sequencing**. Genomic DNA from tissue and blood was extracted using a QIAamp DNA kit (Qiagen, 51306). Whole-exome libraries were generated using an Agilent Sure Select Human All Exon V6 Kit (Agilent Technologies, 5190-8865) following the manufacturer's recommendations. A 0.6 µg aliquot of DNA was used as input material for DNA sample preparation. In brief, fragmentation was carried out with a Covaris LE220 sonicator to generate 180–280 bp fragments. After end repair, adenylation and ligation to adapter oligonucleotides, DNA fragments with ligated adapters on both ends were selectively enriched by PCR. Then, libraries were hybridized with a biotin-labeled probe to capture the exons of genes with streptavidin-coated magnetic beads. The captured libraries were enriched by PCR. Products were purified using an AMPure XP system (Beckman Coulter, A63882) and quantified using the Agilent High Sensitivity DNA assay in an Agilent Bioanalyzer 2100 system. The DNA libraries were sequenced on the Illumina HiSeq platform aiming for 200X coverage for blood as well as tissue, and 150 bp paired-end reads were generated.

**Somatic mutation calling**. DNA sequencing data were processed according to the Genome Analysis Toolkit (GATK, https://software.broadinstitute.org/gatk/) best practices workflow. First, raw data from both WGS and WES were processed with Trimmomatic[59] for adapter trimming and low-quality read filtering. Clean reads were then aligned to the hg19 human reference genome using BWA-mem[60]. SAMtools[61] was used to convert the resulting SAM files to BAM files and then sort the BAM files. PCR duplicate marking was performed with Picard, and base quality scores were recalibrated using the BaseRecalibrator tool in GATK (version 4.0.11.0). Next, Mutect2 was run to call somatic mutations from the tumor-normal paired BAM files. In addition, each normal file was processed with the tumor-only mode in Mutect2, and a panel of normal (PON) files was then generated to filter out expected artifacts and germline variations. For the organoid without a paired blood DNA sample (CAS-NEN-2), only the PON file was used as a control. The resulting VCF files were annotated with ANNOVAR[62], and variations with allele frequencies of less than 0.05 were filtered out.

**Copy number analysis**. For DNA sequencing data, BAM files were processed with CNVKit[63] to call copy number variations, and GISTIC2.0[64] was then used to identify regions of focal gain and loss. Segments of PDPCO and TCGA PAAD cohort[23] data were displayed using Integrated Genomics Viewer (IGV)[65]. Circos plots of CNVs between paired organoid and tissue samples were drawn with the Rcircos package[66] using the data from the "all_thresholded.by_genes.txt" file generated with GISTIC2.0 as CNV scores.

**Mutational signature analysis**. Somatic base substitutions were classified into 96 mutation trinucleotides and the somatic mutation rate of each type of substitutions was calculated to generate a context-specific mutation profile. Mutational signatures were extracted from the context-specific mutation profile by using NMF algorithm ("MutationalPatterns" package). Consensus clustering ("CancerSubtypes" package; clusterAlg = "hc", distance = "pearson", innerLinkage = "ward.D2") was performed on contribution of each mutational signature among all samples.

**Assessment of tumor cell purity**. For whole-exome sequenced data of organoids, tumor purity was estimated with ABSOLUTE (https://software.broadinstitute.org/cancer/cga/absolute). For whole-genome sequenced data of tissues, tumor cell purity was inferred by Sequenza. Concordance represented shared proportion of somatic mutations in coding regions between matched tissue and organoid. Tissue found in organoid represented percentage of the primary tumor mutations found in paired organoid.

**RNA library preparation and sequencing**. PDPCOs and NPOs in Matrigel were collected and washed with precooled PBS before lysis with 1 ml of TRIzol (Invitrogen, 15596018), and total RNA was extracted according to the manufacturer's instructions. A total amount of 3 µg RNA per sample was used as input material for RNA sample preparation. Ribosomal RNA was removed with a Globin-Zero Gold rRNA Removal Kit (Illumina, E7750X), and rRNA-free residue was cleaned up by ethanol precipitation. RNA-seq libraries were generated using a NEBNext Ultra RNA Library Prep Kit (New England Biolabs, E7530L) according to the manufacturer's instructions. After cluster generation, the libraries were sequenced on the Illumina HiSeq X Ten platform, and 150 bp paired-end reads were generated.

**RNA-seq data processing**. We used TopHat[67] to map the sequencing reads of 84 PDPCOs (75 CAS-DACs, 4 CAS-IPMNs, 1 CAS-ACC and 4 CAS-NENs) and 3 NPOs to the human reference genome hg19. Then, fragments per kilobase of exon per million mapped reads (FPKM) values were determined using Cufflinks[68] as the gene expression measurements. Genes with FPKM values <1 in all 87 samples were filtered out. Finally, 15,819 protein-coding genes remained for further analysis.

**PCA and NMF clustering of PDPCOs in the discovery cohort**. We performed PCA based on the expression data of the 2,000 most variable genes across 45 PDPCOs (39 CAS-DACs, 1 CAS-IPMN, 1 CAS-ACC and 4 CAS-NENs) and 3 NPOs. The first two principal components are shown in Fig. 3a. PDPCO and NPO samples were clearly separable by the first two principal components. We further performed NMF on 40 exocrine PDPCOs without sample CAS-ACC. As shown in Supplementary Fig. 3a, the top 2 clusters with the highest cophenetic correlations were n = 3 and n = 4. We selected the model with n = 4, which produced more meaningful results with clear biological significance. Via this analysis, 4 subtypes were identified: class 1, class 2, class 3, and class 4. NMF was performed with the R package NMF[69].

**Biological characteristics of transcriptomic subtypes in the discovery cohort**. We first identified DEGs across the 4 subtypes as genes with an average FPKM value >1 and with a fold change >1.5, a t-test P value <0.05 compared with other samples. All DEGs are shown in the heatmap in Fig. 3b, which was generated using the pheatmap R package[70]. Then, we adopted the transcriptomic signatures from published studies. The basal-like and classical signatures were defined by the union of signatures in previous studies[4,8]. The progenitor, ADEX and immunogenic signatures were also previously defined[3]. GSEA was performed using the JAVA

GSEA_4.0.2 program[71] with a pre-ranked gene list with 1,000 permutations. We used Enrichr to analyze the functional pathways for class 4, which was not enriched in all signatures[72]. We extracted the genes composing the glycolysis pathway and show them in Fig. 3e.

**Validation of transcriptomic subtypes in an internal cohort and a public database**. We used the DEGs of the 4 subtypes (classical-like, basal-like, classical-progenitor, and glycomet) to independently identify stable sample clusters of the remaining 39 exocrine PDPCOs (36 CAS-DACs and 3 CAS-IPMNs) and the TCGA PAAD cohort[23], for which RNA-seq data were collected from the cBio-Portal database[73] with the cgdsr R package[74]. NMF was employed to analyze the RNA-seq data of the 39 exocrine PDPCOs and 165 TCGA PAAD samples. Three and four subtypes were identified in the internal cohort and public database, respectively, based on the results of Fisher's exact test with the 4 defined subtypes. The overlapping DEGs between the discovery cohort subtypes and internal cohort subtypes as well as the TCGA PAAD subtypes are shown in Supplementary Fig. 3b and Fig. 3f. Survival analysis was performed using survival R package and survminer R package on the TCGA PAAD samples with available follow-up data ($n = 157$)[23]. Kaplan-Meier curves of overall survival for patients stratified by the four subtypes, by classical and basal-like subtypes, and by classical-progenitor and other subtypes, are shown separately in Fig. 3g and Supplementary Fig. 3c, d, respectively. Significance was assessed by using log-rank test.

**ATAC-seq library preparation and sequencing**. For each PDPCO line, we prepared two sequencing libraries (technical replicates). PDPCOs in Matrigel were collected and washed with precooled PBS. Cells were washed once with 200 μl of cold PBS buffer and were then centrifuged for 5 min at 500× $g$ and 4 °C. The supernatant was removed and discarded. The cell pellet was resuspended in 50 μl of cold lysis buffer (10 mM Tris-HCl (pH 7.4), 10 mM NaCl, 3 mM MgCl2, 0.5% NP-40) by gently pipetting up and down. The cells were immediately centrifuged at 500Xg for 5 min at 4 °C to collect nuclei, and the transposition reaction was immediately continued. ATAC-seq libraries were generated using a TruePrep DNA Library Prep Kit V2 for Illumina (Vazyme, TD501) according to the manufacturer's instructions. After cluster generation, the libraries were sequenced on the Illumina Nova platform, and 150 bp paired-end reads were generated. Libraries that contained less than 15 million aligned, deduplicated reads were sequenced again, and the additional reads were pooled prior to deduplication.

**ATAC-seq data processing**. Illumina adapter sequences as well as transposase sequences were trimmed from paired-end ATAC-seq reads with a customized script, and the reads were then mapped to the human reference genome hg19 with Bowtie v1.0.0[75]. Duplicate reads were removed with SAMtools v0.1.19[61]. Only uniquely aligned reads were used for peak calling with Hotspot using default parameters (http://www.uwencode.org/proj/hotspot/). HOTSPOT analysis generated two types of peaks: narrow peaks and hotspot regions (broad peaks). In this study, we used the narrow peaks for subsequent analysis.

**ATAC-seq data quality control**. The insert size was directly calculated from the BAM file with SAMtools. Two patterns of the insert size distribution suggested the high quality of the libraries. First, a large proportion of reads with less than 100 bp represented the nucleosome-free region. Second, the fragment size distribution had a clear periodicity indicative of nucleosome binding patterns. To obtain the quality control score (QC score), each TSS was extended to ±2000 bp and overlapped with the insertions. To normalize this value to the local background, the insertions were also overlapped with the region ±2000–3000 bp from the TSS. The final QC score was defined by the ratio between the overlap of the foreground and the overlap of the background.

**Quantification of chromatin accessibility with the openness score**. ATAC-seq can measure the accessibility of chromatin in the region represented by a given peak. We quantified the openness of ATAC-seq peaks by a simple fold change score, which calculated the enrichment of read counts by comparing the peak with a large background region. In brief, let N be the number of reads in peaks of width L and G be the number of reads in the background window W (1 Mb in our study) around this peak. The openness score of the peak O can thus be defined as follows:

$$O = (N/L)/(G/W) \quad (1)$$

**Identification of reproducible peaks**. For each technical replicate, we selected the one with the higher QC score to quantify the chromatin accessibility of this sample. We first filtered out the peaks called in a single sample and overlapped with the ENCODE hg19 blacklist (https://www.encodeproject.org/annotations/ENCSR636HFF/) to remove artifacts.

We further processed the overlapping peaks across samples using an iterative removal procedure. The most significant peak was kept, and any peak that directly overlapped with that significant peak was removed. Then, this process was iterated for the next most significant peak until all peaks had either been kept or removed.

ATAC-seq track visualization was performed with the WashU epigenome browser[76].

Finally, we defined the reproducible peaks based on the ATAC-seq data of 41 exocrine PDPCOs (39 CAS-DACs, 1 CAS-IPMN and 1 CAS-ACC) and 4 neuroendocrine PDPCOs (CAS-NEN-1, CAS-NEN-2, CAS-NEN-3, CAS-NEN-4). To increase credibility, the most accessible 30,000 peaks with a length ≥300 bp in each sample were used, among which peaks observed in at least 2 samples were labeled reproducible peaks, constituting a high-quality reproducible peak set for exocrine PDPCOs (156,721) and neuroendocrine PDPCOs (26,975). For comparison with published pan-cancer data, we generated a PDPCO-wide reproducible peak set by directly merging the peak sets of the exocrine and neuroendocrine PDPCOs. For comparison of reproducible peaks across samples, we established an openness score matrix and performed normalization with a quantile normalization strategy using the normalize.quantiles function in the R package preprocessCore[77].

**Chromatin state analysis of reproducible exocrine PDPCO and pancancer peaks with chromHMM**. To investigate the distribution of exocrine reproducible PDPCO peaks in the chromatin states and compare these data with pan-cancer data, we used ChIP-seq-defined chromHMM states from the Roadmap Epigenomics Project. ChromHMM 15 state models were downloaded from the chromatin state learning site (https://egg2.wustl.edu/roadmap/web_portal/chr_state_learning.html). We then determined the number of regions of each chromHMM state model were overlapped by ATAC-seq peak midpoints. To determine the significance of these overlaps for each chromHMM state, we compared the proportion of ATAC-seq midpoints overlapping the given chromHMM state with the expected background determined from the total length covered by the chromHMM state and the length of the hg19 genome via a binomial test in R. The genomic locations were annotated with the annotatr R package[78].

**Principal coordinates analysis of reproducible PDPCO and pan-cancer peaks**. To investigate the similarity between the reproducible peaks in the PDPCO and pan-cancer data, we performed principal coordinates analysis, which takes a set of dissimilarities or distances as input and returns a set of points. The distances between the points are approximately equal to the dissimilarities. We first evaluated the similarity by computing the Jaccard similarity. Then, we created the distance matrix by subtracting the similarity values from one and took this matrix as input. The number of dimensions was set to 2, as we considered only the first and second coordinates.

**Inference of TF regulons from RNA-seq data**. We used the SCENIC workflow[79] to identify the gene regulatory network (TF regulon) and scored the activity of the TF regulon. In brief, SCENIC was first used to identify the potential TF-target regulatory links according to the respective relevance of TFs for the prediction of the expression of target genes (TGs). Then, direct TF-target regulons were identified by motif binding patterns around the TSS. The regulon activity score for a single sample was finally calculated by computing the AUC. We first inferred the regulons and their activity by taking the gene expression data of the 41 exocrine PDPCOs and 4 neuroendocrine PDPCOs as input. To identify the specific regulon, we performed differential analysis based on the regulon activity score and calculated the adjusted P-value for each regulon. Differential analysis was applied to exocrine and neuroendocrine tumors, as well as one and others for the 4 PDAC subtypes.

**Motif enrichment analysis of ATAC-seq data**. To identify the key regulators for specific peaks, we performed motif enrichment with HOMER[80]. For comparison between NEN and PDAC, we used the merged NEN and PDAC peaks as input, while for comparison among PDAC subtypes, we used the differential peaks. We ranked the TFs ($TF_1, TF_2, \ldots, TF_N$) according to the P-values associated with one-sided unpaired $t$ test adjusted for multiple testing using false discovery rate (FDR), denoted as $R_i^j$ for the $ith$ TF in the $jth$ subtype. Then, we used the difference between the average rank for other subtypes and the rank for the subtype of interest ($S_i^j$) to measure the relative enrichment levels of motifs:

$$S_i^j = \frac{1}{J-1} \Sigma_{k \neq j}(R_i^k - R_i^j) \quad (2)$$

where $S_i^j$ is the relative enrichment score, and a positive score suggests that motif $i$ is enriched in subtype $j$. $J$ is the number of subtypes, which was set to 2 for comparison between the exocrine and neuroendocrine tumors and 4 for comparisons between PDAC subtypes.

**Identification of subtype-specific regulons by integrating RNA-seq and ATAC-seq data**. We inferred 377 TF regulons from gene expression profiles. 283 of which were available for motif enrichment analysis with HOMER (Supplementary Data 4). We next identified subtype-specific regulons among the 283 TF regulons.

For TF $i$ in a given subtype $j$, we defined the specific score $SS_{i,j}$ as follows,

$$SS_{i,j} = \begin{cases} -\log_{10} p^1_{i,j} & if \ p^1_{i,j} < 0.05 \ and \ S^j_i > 0 \\ -\log_{10} p^1_{i,j} & if \ p^1_{i,j} < 0.05 \ and \ p^2_{i,j} < 0.05 \\ 0 & otherwise \end{cases} \quad (3)$$

where $p^1_{i,j}$ is the $P$ value from the $t$-test of the regulon activity score and $p^2_{i,j}$ is the $P$ value from motif enrichment analysis as described above. $S^j_i$ is the relative enrichment score, and a positive score suggests that motif $i$ is enriched in subtype $j$.

The subtype-specific regulon set $A_j$ in subtype $j$ was composed by all the TFs with positive specific score $SS_{i,j}$ as follows (a $t$-test $P$ value < 0.05 as well as a relative enrichment score > 0 or motif enrichment $P$ value <0.05 for TF $i$ in subtype $j$).

$$A_j = \{TF_i | SS_{i,j} > 0\} \quad (4)$$

**Links between ATAC-seq peak openness and targeted gene expression**. To identify putative regulatory links between ATAC-seq peak accessibility and gene expression, we correlated the ATAC-seq and RNA-seq data across 41 exocrine PDPCOs (39 CAS-DACs, 1 CAS-IPMN and 1 CAS-ACC). To distinguish links between promoters and distal elements, we annotated the peaks overlapping with the region 2 kb upstream of the TSS as promoters and the other peaks as distal elements. We next performed the same method to predict links on the two types of peaks. First, all possible interactions between ATAC-seq peaks and genes within 500 kb of the peaks were identified. Then, we removed the bottom 25% of both the genes and peaks based on variance. For each of these interactions, we calculated the Pearson correlation coefficient and obtained the P-value for the comparison between the openness score of the ATAC-seq peak ($\log_2$(openness+1)) and the expression level of the gene ($\log_2$(FPKM + 1)). $P$ values associated with Pearson's correlation coefficients were adjusted for multiple testing using FDR. Positive links were defined by adjusted P-value <0.25 and Pearson correlation coefficient >0.5, and negative links were defined by adjusted $P$ value <0.25 and Pearson correlation coefficient <−0.5. Promoter-to-gene links contained only positive correlations, while distal element-to-gene links consisted of both positive and negative correlations.

**Exploration of potential functional noncoding mutations**. To explore potential functional noncoding mutations, we used a fold change-based approach. First, we included the noncoding mutations located within peaks; thus, 7,398 mutations were retained. We then calculated the fold change FC in the peak openness at the mutation site as follows:

$$FC = O_m/O_w \quad (5)$$

Where $O_m$ is the average peak openness score of mutated PDPCOs and $O_w$ is the average peak openness score of wild-type PDPCOs. In total, we gained 547 upregulating mutations with fold change >1.5 and 969 downregulating mutations with fold change < 0.67, as shown in Fig. 5c. For survival analysis, Kaplan-Meier survival analysis with survival R package and survminer R package was performed on TCGA PAAD samples with available patient follow-up data ($n = 157$)[23]. Patients were sorted by corresponding gene expression levels, and we compared the top 33% (high) with the bottom 33% (low). P-values were determined by using log-rank test.

**High-throughput screening of chemical and chemotherapeutic drugs**. White, clear bottom 384-well plates were coated with 10 µl of collagen at room temperature for at least one hour using a Multidrop Combi reagent dispenser before the addition of organoid suspensions. PDPCOs were dissociated with Tryp-LE before being resuspended in medium and dispensed into 384-well plates (3,500 cells per well). The next day, 283 compounds (Selleck), as well as DMSO controls were added in duplicate using a Bravo robotic workstation. To assess cell viability, 25 µl of CellTiter-Glo Reagent per well was added after three days. The plates were gently shaken for 15 min at room temperature before luminescence was assessed using an Envision plate reader. Average inhibition rates from two independent experiments were calculated with Excel and visualized using GraphPad Prism 8. Detailed information of involved reagents was provided in Supplementary Data 11.

Five chemotherapeutic agents (gemcitabine, paclitaxel, 5-fluorouracil, oxaliplatin and irinotecan) and chemicals that had significant inhibitory effects on cells were used for the secondary screening. The range of concentrations selected for each chemical was based on the primary screening data (Supplementary Data 7). Organoids were similarly dispensed into 384-well plates. Concentration dilution and addition of each compound were performed with the Bravo robotic workstation. Cell viability was assessed using CellTiter-Glo Reagent after three days of incubation with drugs. The secondary screening was performed in technical duplicate (same screening run), and all screening plates were subjected to stringent quality control measures. To measure sensitivity, we used 5-point dose-response curves; for each drug, was 5 concentrations and the corresponding cell viability values were used as input for curve generation. The viability was set to 100 if it was higher than baseline, and each drug concentration (nmol/L) was log10 transformed. The AUC was calculated with the sintegral function in R, and the normalized AUC was obtained by dividing one AUC by the maximum AUC for each drug. The AUC heatmap for the secondary screening was generated with GraphPad Prism 8.

**Links between ATAC-seq peak openness and drug sensitivity**. To identify putative links between ATAC-seq peak accessibility and drug sensitivity across 39 exocrine PDPCOs (38 CAS-DAC and 1 CAS-IPMN), we performed Pearson correlation analysis. The $P$ value between the ATAC-seq peak openness score ($\log_2$(openness+1)) and the drug AUC was calculated. Positive links were defined by $P$ value <0.01 and Pearson correlation coefficient >0.5, while negative links were defined by $P$ value <0.01 and Pearson correlation coefficient <−0.5. In subsequent analysis, samples were sorted by the peak openness score. Gene expression and drug sensitivity were compared separately among representative samples and all samples by a one-tailed t-test.

**Clinical follow-up and recurrence-free survival analysis**. Clinical follow-up data were collected from the Changhai Hospital prospective database. Patients regularly received tumor marker and imaging examinations every three months after surgery and were followed up at the same time interval by outpatient clinic visits and/or telephone contact. Recurrence was confirmed by imaging examination, including contrast-enhanced computed tomography or contrast-enhanced magnetic resonance imaging. Recurrence-free survival curves and tumor marker curves were visualized using GraphPad Prism 8. $P$ value for Kaplan-Meier survival analysis of the recurrence-free survival was determined by using log-rank test.

**In vivo experiments**. All In vivo experiments were performed according to the Institutional Animal Care and Use Committee (IACUC) of the Center for Excellence in Molecular Cell Science (CEMCS), and ethical approval was received from the IACUC of CEMCS. PDPCOs were suspended in medium containing 50% Matrigel and injected subcutaneously into 8-week-old female SCID mice ($2 \times 10^6$ cells/injection), which were maintained according to Shanghai Laboratory Animal Center Institutional Animal Regulations (SPF mouse room with 12 h of light, a temperature of 18–23 degrees Celsius, and a humidity of 40–60%). Mice were checked for tumor development every two or four days. For histological and immunohistochemical assessment of the ODXs, mice were sacrificed for xenograft harvesting when the tumor diameter reached approximately 0.5 cm. For drug susceptibility evaluation of the ODXs, when the average tumor diameter reached approximately 0.5–0.6 cm, mice were treated with either 5-fluorouracil (25 mg/kg), GEM (50 mg/kg), PTX (3.5 mg/kg) or Vehicle. The CAS-DAC-18, CAS-DAC-20, CAS-DAC-22 and CAS-DAC-24 tumors were treated with drugs and assessed with Vernier calipers every 2 days, while the CAS-DAC-36 tumor was treated every 3 days. After 2 weeks (CAS-DAC-18, CAS-DAC-20, CAS-DAC-22 and CAS-DAC-24) or 4 weeks (CAS-DAC-36) of treatment, mice were sacrificed for xenograft harvesting. The ethically approved maximal tumor size of 15 mm by the IACUC of CEMCS was not exceeded in this study. The tumor volume V (mm³) was calculated with following equation.

$$V = 0.5 \times length \times width^2 \quad (6)$$

**Statistics & Reproducibility**. All statistical calculations were implemented in R (version 3.6.3; https://cran.r-project.org/). The detailed statistical tests were indicated in figures or associated legends where applicable. No statistical method was used to predetermine sample size. No data were excluded from the analyses. Mice used in our study were randomly divided into different groups at the same age. The Investigators were not blinded to allocation during experiments and outcome assessment.

**Reporting summary**. Further information on research design is available in the Nature Research Reporting Summary linked to this article.

## Data availability

All raw sequencing data has been deposited in Genome Sequence Archive (GSA) database that is publicly accessible with accession number HRA002013. As consent was only provided for data use for non-commercial purposes, the data will be available via a materials transfer agreement to any non-commercial parties. Access will be granted and the data can be downloaded in a typical one month time window. The public dataset (TCGA PAAD cohort) is available at cBioPortal [https://www.cbioportal.org/]. All other data can be found within the main manuscript or the source data file. Source data are provided with this paper.

## Code availability

The custom code used is available at GitHub [https://github.com/amssyqy/Pancreatic-cancer-organoids/tree/v1.0.0][81].

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

## Acknowledgements

We would like Shujue Lan, Yunqing Ci and Ming Chen for technical help at the SIBCB Core Facility. This study was supported by grants from the Strategic Priority Research Program of the Chinese Academy of Sciences (Nos. XDA16020905 to D.G. and XDB38040400 to L.N.C.), the National key research and development program of China (Nos. 2017YFA0505500 to D.G. and L.N.C., 2020YFA0509000 to D.G. and 2020YFA0712402 to Y.W.), the Basic Frontier Science Research Program of Chinese Academy of Sciences (No. ZDBS-LY-SM015 to D.G.), the National Natural Science Foundation of China (Nos. 32125013 to D.G., 81830054 to D.G., 81772723 to D.G., 31930022 to L.N.C., 31771476 to L.N.C., 12131020 to L.N.C., 12026608 to L.N.C., 82002559 to S.Z.G, 82172589 to S.W.G., 81972913 to G.J., 82172712 to G.J., 12025107 to Y.W., 11871463 to Y.W., and 61621003 to Y.W.), the Shanghai Science and Technology Committee (Nos. 21XD1424200 to D.G., 21ZR1470100 to D.G., 20ZR1456500 to S.W.G. and 20511101200 to G.J.), JST Moonshot R&D (No. JPMJMS2021 to L.N.C.), and the Shanghai ShenKang hospital development center (No. SHDC2020CR2001A to G.J.).

## Author contributions

D.G. and G.J. conceived and designed the experimental approach. X.H.S. and Y.G.L. performed most experiments. Q.Y.Y., S.J.T., S.W.G., Y.W., and L.N.C. contributed to the computational analysis and statistical analysis. Y.H.Z., J.H., X.Y.Z., M.H., Z.L., Y.Q.Z., S.Z.G., H.W., X.F.X., K.L.Z., and W.J. helped the experiments and provided technical support. L.N.C., Y.W., G.J., and D.G. prepared the manuscript as senior authors.

## Competing interests

The authors declare no competing interests.
