## [Peer Review File · Nature Communications]

Integrated profiling of human pancreatic cancer organoids reveals chromatin accessibility features associated with drug sensitivityREVIEWER COMMENTS

Reviewer #1, expert in organoids (Remarks to the Author):

Shi X et al. established a PDAC organoid biobank with comprehensive molecular analysis and drug screening. This paper is well written and highly informative for cancer researchers. In addition, they found enhancer mutations and epigenetic lesions associated with drug susceptibility. Altogether, I would like to recommend its publication. Some concerns need to be addressed. I also suggested several points that could improve the quality of papers, as follows.

1. In figure2, they showed driver gene mutations of PDAC organoids in WGS data. However, some important genes were lacking in the list, such as KDM6A, BRCA1/2, RB1 (for NEC), and MEN1 (for NET). In addition, the only list of non-coding gene mutations was shown in supplementary tables.

2. Gene mutation signatures for PDAC organoids should be shown.

3. Identification of Glycomet subtype is interesting. Was this subtype associated with genetic mutations and chromosomal instability?

4. In Fig3g, the prognosis data is discrepant with the previous report. The authors should confirm if the previous subtype classifications (Basal vs. Classical, Bailey's classification) can detect the previous papers' differences.

5. NENs and ACC organoids had a distinct transcriptome profile in Fig. 3a. It would be great if the authors could show representative gene expressions of these organoids, such as acinar cell markers and neuroendocrine markers.

6. The authors showed the association between chromatin accessibility peak and drug sensitivity. This data is interesting, but it is uncertain if these associations are specific or a random effect. Which gene was associated with each peak shown in Fig5h, i?
NCOR gene regulates histone deacetylation other than JAK/STAT pathway. The authors should check if the Go6976 sensitive PDAC organoids have phosphor STAT3 upregulation. Similarly, they need to check the deacetylation status of MSH2 in Panobinostat sensitive PDAC organoids.

7. How accurately the high throughput analysis was carried out. Z' factor?

Is there any known gene-drug correlation, such as TP53 mutation vs Nutlin-3.

AUC value/drug/line (Extended Data Fig5 is hardly visible)
should be shown in supplementary table.

In Fig6g, they showed the consistent drug responses in xenografts. CAS-DAC-36 showed slow tumor growth, and it is difficult to judge if they were resistant to chemotherapy or did not grow in the mice. Perhaps, the authors needed to observe for a longer duration when tumors grow slowly.

Reviewer #2, expert in ATAC-seq analysis (Remarks to the Author):

In the manuscript, "Integrated profiling of human pancreatic cancer organoids reveals chromatin accessibility to drug sensitivity networks" Shi and colleagues generated 84 pancreatic cancer organoid lines, termed patient-derived pancreatic cancer organoids (PDPCO), from pancreatic cancer patients primary tumor, 80 from exocrine pancreatic tumors and 4 from neuroendocrine tumors. The authors have also generated 6 normal pancreatic ductal organoids as a control. The authors then assayed the genomic, chromatin accessibility landscape and transcriptome in those organoids by performing WGS, ATAC-seq and RNA-seq. The authors used the WGS to call known and unknown mutations from the PDPCO and compare the PDPCO mutations to matching pancreatic tissue whole exome sequencing to show the similarities between tissues and organoid. Using RNA-seq transcriptome data and ATAC-seq they define 4 different subtypes of pancreatic cancer consisting of 2 well known subtypes (basal-like and classical) and 2 new subtypes that are defined by the authors as classical-progenitor and glycomet. Next the authors use ATAC-seq and RNA-seq data together to better define regulatory landscape, by relying on SCENIC regulon

analysis they define RNA to ATAC connectivity and using ATAC/RNA they show TF enrichment in the different subclasses of pancreatic cancer and define pancreatic cancer gene-to-peak matrix. The author then goes into the combination of all data types they generate in order to get links between genomic mutations ATAC-seq peaks and RNA expression in pancreatic cancer.

We find the paper "Integrated profiling of human pancreatic cancer organoids reveals chromatin accessibility to drug sensitivity networks" could be an important contribution to the field both as a great resource for the pancreatic cancer community and as an source of information for new transcription factor importance and mutations and their importance to different ATAC-seq peaks and their linkage to genes.

In the abstract the authors list the following claims:

Identified distinct transcription factors that distinguish molecular subtypes of pancreatic cancer
predicted numerous chromatin accessibility peaks associated with gene regulatory networks
discovered novel regulatory noncoding mutations with potential as cancer drivers
revealed the chromatin accessibility signatures associated with drug sensitivity

Major Issues :

HOMER motif score and motif enrichments. Figure 4 B/C shows motif enrichments calculated using rank TF lists as described in the methods. We wonder why not display HOMER enrichment from each cluster while using a background list of all peaks detected. This way we will have strong TF enrichments in each cluster over the genome and not a list of TFs that are enriched between clusters.

We find the text description of Figure4 B/C confusing, while the data seen in Fig4 B/C seems to come from HOMER TF enrichment ($S_{i,j}$ in the methods) it is not described in the text and we find it confusing with the RNA/ATAC score ($SS_{i,j}$) in methods.

In their drug screen analysis, the authors use an approach that correlates cell drug sensitivity and ATAC-seq peaks. This approach limits the author resolution in ATAC-seq space and leads them to two classes drug sensitive/non-sensitive (as seen in Figure5G). This results worries us that most of the changes seen in the results is due to cell death/apoptosis and cell stress in response to the drug. We think there is a place here for a more cluster based analysis of only the ATAC-seq peaks, this analysis will give more emphasis to changes in ATAC-seq and will allow TF enrichment analysis, following this the authors can test which drugs are more correlated with the peaks change in each cluster.

Minor Issues :

WGS comparison between organoids and tissue: The authors choose to show the similarities in genomic scale but both class specification and gene level analysis (Extended figure 2B and 2C) show differences between the organoids to the tissue. It will be useful to have a more accurate analysis showing the similarities and changes between the tissues to organoids.

The authors claim 527,272 new non coding mutations and supply a detailed table where one can generate all the info but some information about the mutations and how many of them are only found in one sample and how many are shared is missing.

The authors define 4 sub class of pancreatic cancer claiming that 2 of the subtypes are new, the authors fail to compare their new subtypes to many of the proposed subtype definitions that are found in the literature, for a nice summary of the different classifications (30419209)

Extended Data Fig. 4 panel b and c are in reverse order in the legend
Page 7 Line 21, Sends reader to Fig4a while meant 4b

Reviewer #3, expert in multi-omics and drug response (Remarks to the Author):

Shi et al. provided an overview of the chromatin accessibility profiles of 84 patient-derived pancreatic cancer organoid lines (PDCPOs); and integrated it with paired whole-genome, transcriptome and drug screening for 283 epigenetic-related chemicals and 5 chemotherapy agents. The authors reported that the histological and morphological patterns, as well as the exome landscape (specifically, cancer specific copy-number and mutational alterations) of the primary pancreatic tumors were conserved in the established PDCPOs. Unsupervised clustering of the transcriptomic profiles revealed a novel glycometabolic subtype within pancreatic ductal adenocarcinomas (PDAC), however this subtype did not yield distinct prognostic stratification compared to known subtypes (classical-like and basal-like). Next, the authors performed a descriptive analysis where they integrated the chromatin accessibility with the transcriptome data to identify subtype specific transcription factor regulons in PDAC; and putative in-cis 'chromatin-accessibility-peak to gene-expression' links. By linking the genomic profiles at these identified chromatin-accessibility peaks, the authors also proposed regulatory noncoding mutations with potential cancer-related functional consequences. Next, correlating the drug sensitivity data with the chromatin accessibility peaks yielded potential biomarkers for predicting therapy resistance or sensitivity for epigenetic-related chemicals and chemotherapy agents such as 5-fluorouracil and paclitaxel. Finally, the authors reported that patients with paired organoids demonstrating sensitivity to chemotherapy agents survived significantly longer post-surgery and adjuvant therapy. Post-treatment imaging and cancer-antigen marker data available for the patients, as well as drug screening data in paired organoid-derived xenografts (generated by the authors) further corroborated the PDCPO drug response thus showcasing its utility in personalized medicine.

Synopsis:

The authors establish patient-derived pancreatic cancer organoid lines and performed an integrative analysis of paired chromatin accessibility, transcriptomic, genomic profiles and high-throughput drug screening. Overall, the study provides a useful organoid with some novel findings that demonstrate the utility of organoids in predicting therapeutic outcomes in pancreatic cancer. However, the rationale for some of the approaches is not clear and impacts the ability to assess the significance of the relationships highlighted and conclusions drawn from them.

Additional investigations are suggested as follows.

1. Tissue microenvironment can contribute to a large confounding effect in cancer epigenomic studies due to different cell types have distinct epigenetic signatures. The authors have indicated that analyzing ATAC-seq profiles in primary pancreatic tumor tissue can be challenging due to presence of excessive stroma. The authors should report on the (microenvironment) composition of the patient-derived organoid lines generated compared to the primary tumors, and state what measures were taken place to account for any confounding effects (if present).
2. The authors should clarify how many tests were performed to identify subtype-specific TF regulons (for $p_{(i,j)}^1$ and for $p_{(i,j)}^2$). For a large number of tests, the authors should use FDR (or similar) adjusted p-values.
3. The authors calculate correlations over all possible interactions between ATAC-seq peak accessibility and gene expression within 500 kb of the peaks (after filtering for low variance) to identify putative links. The authors should state the total number of interactions tested and use FDR (or similar) adjusted p-values.
4. The authors calculate correlations over possible interactions between ATAC-seq peak accessibility and drug sensitivity to identify putative links. The authors should state the total number of interactions tested and use FDR (or similar) adjusted p-values.
5. In Figure 5j, 10 representative samples were chosen to showcase differential accessibility at a putative enhancer locus. Could the authors clarify how these samples were chosen? Were these the samples that had the most extreme differences?
The authors should repeat the boxplots Figure 5k,l,m over all samples (if necessary stratify the ATAC-peak into tertiles – high, medium, low).

Could the authors also clarify/ repeat this for Extended Data Figures 5g-q.

6. The authors have proposed specific examples of putative chromatin accessibility peaks as predictive biomarkers of drug response in the PDCPOs. Can the authors comment on whether these epi(genomic) alterations presented intrinsic resistance/ sensitivity or could they be inducing resistance to the therapy. And specifically discuss their application in novel treatment strategies?

7. The authors have compiled and performed drug screening on a large library of chemicals targeting epigenetic-related signaling pathways, with a decent number (59) showing inhibition in 1 or more PDCPOs. The authors should provide a figure for the resistance/ sensitivity for these agents in the PDCPOs similar to Figure 6a.

8. Do the authors have any data on the consequence of any epigenetic-related therapies on the epigenome of the PDCPOs? For instance, ATAC-seq of the PDCPOs before and after treatment for epigenetic-therapies demonstrating strong inhibition.

9. Were post treatment imaging results available for more than 3 patients: CAS-DAC-24/22/20? If so, the authors should provide a graphical summary of the correlation between the imaging results and organoid sensitivity for all patients with available paired data treated with the same (or similar mode-of-action) chemotherapy agents as they have done for these 3 examples.

Could the authors improve the clarity for the approaches used and conclusions drawn for the following?

1. What was the rationale to use $\log_2(\text{copy number})/2 > 0.1$ as a threshold for gains/ amplifications in the Extended Data Figure 2. Would $\log_2(\text{copy number}/2) > 0.1$ be a better threshold?

2. The authors indicate that the somatic mutational profiles of the PDCPOs were similar to that observed in the TCGA pancreatic adenocarcinoma (PAAD) dataset. Could the authors include evidence for this conclusion?

3. Line 1-2 on page 6 is unclear. Can the authors consider rephrasing?

4. Extended Data Figure 3a details multiple solutions for determining the number of clusters for NMF of the transcriptomic profiles for the exocrine group. The authors should clarify why $k=4$ was chosen over the other solutions? Also check the legend for $k=3,4$ in the figure.

5. In Figure 3b/ Extended Data Figure 3b, the heatmap columns are indicated by samples, however, the legend states otherwise.

6. Figure 4b is not mentioned in the text.

7. In line 27, page 8 the authors state that "95% of genes were regulated by multiple peaks (≤ 6) in the distal ATAC-seq peak to-gene links". However, this conclusion is incorrect since most of these genes were regulated by only 1 peak.

8. The authors should clarify what is the evidence to back the claim "ATAC-seq peak at chr6: 139728666-139729424 predicted sensitivity to oxaliplatin" in line 23, page 6.

9. Figure 6a legend is misleading in how PDCPOs were divided into 3 categories for chemo sensitivity/ resistance. The figure suggests this is done by tertiles, while the text/ figure annotation suggests that is done by AUC value thresholds of 33%,67%. Consider rephrasing.

10. The authors demonstrate CAS-DAC-14, as an example with significant inter-agent variability within a single organoid. The authors should include a figure demonstrating this graphically for all organoids, and link each patient with the 3 groups defined for Figure 6c to ease the understanding of the group definitions. For instance, it is not immediately clear what patients were included in the

resistance group – did these patients demonstrate resistance in the paired organoids for ALL chemotherapy agents?

11. The authors generated organoid derived xenografts (ODX) for 3 patients: CAS-DAC-18/20/36. The authors should clarify whether ODXs were attempted/ generated for other patients and why these 3 were chosen to be described in the manuscript (others had also shown similar resistance/ sensitivity to chemotherapy agents such as CAS-DAC 22/24).

12. What timepoint was used for measuring reduction in tumor growth in the ODXs post treatment. ODX-36 response looks similar at day 20 for all 3 agents shown, but the authors indicate a notable difference. Could the authors clarify this?

Please see below for the detailed responses to each Reviewer's comments

Reviewer #1

Shi X et al. established a PDAC organoid biobank with comprehensive molecular analysis and drug screening. This paper is well written and highly informative for cancer researchers. In addition, they found enhancer mutations and epigenetic lesions associated with drug susceptibility. Altogether, I would like to recommend its publication. Some concerns need to be addressed. I also suggested several points that could improve the quality of papers, as follows.

Point 1: *In figure2, they showed driver gene mutations of PDAC organoids in WGS data. However, some important genes were lacking in the list, such as KDM6A, BRCA1/2, RB1 (for NEC), and MEN1 (for NET). In addition, the only list of non-coding gene mutations was shown in supplementary tables.*

We thank Reviewer #1 for this valuable advice. We summarized the somatic mutation profiles of our PDPCOs, in which top mutated genes of pancreatic ductal adenocarcinoma (PDAC), acinar cell carcinoma (ACC) and intraductal papillary mucinous neoplasms (IPMN) were listed at left panel and those of pancreatic neuroendocrine neoplasms (NEN) were at right (Fig. 2). As suggested, we carefully checked the specific genes (*KDM6A*, *BRAC1/2*, *RB1* and *MEN1*) in the four neuroendocrine tumor-derived organoid lines and showed that they were not mutated. However, among our NEN organoid lines, we have identified many other typical gene alterations. Firstly, *KRAS* mutation and *TP53* mutation, which are common in malignant NENs, were found in CAS-NEN-1 and CAS-NEN-4. Secondly, the G1 phase NET organoid line CAS-NEN-4 with few mutations exhibited significant deep deletion in CNV of *DAXX*, which is closely correlated with pancreatic neuroendocrine tumorigenesis. Another aggressive G2 phase NET organoid line CAS-NEN-2 carried both deep deletion of *DAXX* and *TP53* mutation, which mirrored its clinical metastasis. These results provided further biological identification of these neuroendocrine neoplasm-derived organoids besides for histological examination. Following the advice, we have supplied the detailed information of coding gene mutations in the Supplementary Table 2 in the revised manuscript.

Point 2: *Gene mutation signatures for PDAC organoids should be shown.*

We agree with Reviewer #1's constructive suggestion and perform gene mutational signature analysis in the revised manuscript. Three mutational signatures (A, B and C) were identified by the NMF method in the 70 exocrine organoids, which were highly similar to the Cosmic signatures: Cosmic signature 5, 17 and 1, respectively (Extended Data Fig. 2b). Cosmic signature 1 and 5 were reported to exist in all cancer types, and Cosmic signature 17 was also found in many gastrointestinal cancers. The aetiology of Cosmic signature 5 and 17 remained unknown, while Cosmic signature 1 was thought to be the result of an endogenous mutational process initiated by spontaneous deamination of 5-methylcytosine. Based on signature exposure, the 70 exocrine organoids were divided into three clusters, among which the cluster enriched for signature C accounted for the majority of exocrine organoids (47/70) and signature B exhibited notable advantages in only two organoid lines (2/70) (Extended Data Fig. 2c). These results revealed the commonly enriched gene mutational signatures in exocrine organoids and Cosmic signature 1 as well as Cosmic signature 5 achieved the major compositions. The above contents have been supplied in the revised manuscript.

Point 3: *Identification of Glycomet subtype is interesting. Was this subtype associated with genetic mutations and chromosomal instability?*

We are grateful that Reviewer #1 had such shrewd observation. Actually, we do have carefully investigated the association between RNA subtypes and genetic mutations as well as chromosomal instability, and we would like to share with Reviewer #1 for some of our new data, which are still regarded under the confidential level for both our collaborators and us because we have intended to prepare another publication independently. Chromosomal instability was summarized with a weighted sum approach (Vasaikar et al., Cell, 2019, PMID: 31031003), in which the absolute log₂ copy ratios of all segments within the given chromosome were weighted by the length of each segment and summed up to derive the CIN score. Firstly, the comparison of chromosomal instability showed that there was no significant difference between RNA subtypes (Referee Fig. 1a). Secondly, we explored the correlation of total mutation number, which also exhibited similar results between RNA subtypes (Referee Fig. 1b). Finally, we focused on the specially enriched genetic mutation and CNV in each RNA subtypes. Among all genes, a total of 7 genetic mutations and 26 CNVs were found enriched in one of the RNA subtypes (Referee Fig. 1c). Further sheltered by cancer-related genes, we showed that glycomet subtype was significantly enriched for *SMAD4* mutation (Referee Fig. 1c). Previous studies have demonstrated that *SMAD4* deficiency was responsible for increased aerobic glycolysis in both colon cancer and pancreatic

cancer (Papageorgis et al., Cancer Res, 2011, PMID: 21245094; Liang et al, Gut, 2020, PMID: 31611300), which highly consisted with the biological feature of glycomet subtype. In this study, we mainly focused on identifying the general transcriptomic classification of our PDPCOs rather than an intensive investigation into specific classes.

Referee Fig. 1 | Comparison of genetic mutations and chromosomal instability between RNA subtypes in the 69 exocrine PDPCOs. a, Comparison of chromosomal instability between each two RNA subtypes and differences were calculated by Dunn's multiple comparisons test. **b,** Comparison of total mutation number between each two RNA subtypes and differences were calculated by Dunn's multiple comparisons test. **c,** The specifically enriched genetic mutations and CNVs in one of the RNA subtypes among cancer-related genes (top panel) and all genes (bottom panel). Gene names in red indicated the specifically enriched genetic mutations and CNVs of glycomet subtype. The $-\log_{10}$ (P-value) was showed at right.

Point 4: In Fig3g, the prognosis data is discrepant with the previous report. The authors should confirm if the previous subtype classifications (Basal vs. Classical, Bailey's classification) can detect the previous papers' differences.

We thank Reviewer #1 for pointing out this important concern. In order to examine if the previous subtype classification (Basal vs. Classical, Bailey's classification) can detect the difference on prognosis in previous paper, we generated previous classification on the TCGA PAAD cohort, which was used for survival analysis in our study. According to the signature genes of Bailey's classification (Squamous for Basal and Progenitor for Classical, Referee table 1), we divided the TCGA PAAD cohort clearly into two classes: squamous and progenitor (Referee Fig. 2a). In further survival analysis, better outcomes could

be observed in pancreatic cancer patients classified as progenitor subtype compared with squamous subtype with a borderline p-value of 0.049, which consisted with the prognosis prediction in previous paper (Referee Fig. 2b). In this study, we identified 4 classes to improve the prognostic value of transcriptomic subtype: classical-like, basal-like, classical-progenitor and glycomet, among which traditional classical subtype was subdivided into classical-like class and classical-progenitor class. As expected, traditional classical subtype (classical-like and classical-progenitor) exhibited a trend of better survival in the survival analysis performed with validation in TCGA PAAD dataset, in which no statistical significance might be due to the existence of another glycomet class (Extended Data Fig. 3). More importantly, we revealed that only the classical-progenitor class had a favorable prognosis, while classical-like class showed similar poor outcomes to the other two classes. These results indicated that such transcriptomic subtype not only recapitulated the prognostic value of previous basal and classical classification, but also further unveiled potential survival heterogeneity in conventional classical subtype.

Point 5: NENs and ACC organoids had a distinct transcriptome profile in Fig. 3a. It would be great if the authors could show representative gene expressions of these organoids, such as acinar cell markers and neuroendocrine markers.

We thank Reviewer #1 for the constructive suggestion to help improve our study. According to the previous studies, we identified several representative genes of acinar cell, neuroendocrine cell and PDAC, which were used for comparison. In the ACC organoid line, we found markedly higher expression of acinar cell lineage marker such as *AMY2B* (Park et al., Cancer Res, 2015, PMID: 26122842), *BHLHA15* (also known as *MIST1*) (Direnzo et al., Gastroenterology, 2012, PMID: 22510200), *SLC38A3* (Rooman et al., Pancreatology, 2013, PMID: 24075511) (Extended Data Fig. 3a). For the 4 neuroendocrine organoid lines, significant enrichment of well-known neuroendocrine marker genes like *NKX6-1* (Tseng et al., Am J Surg Pathol, 2015, PMID: 25871618), *NEUROD1* (Bailey et al., Nature, 2016, PMID: 26909576), *NKX2-5* (Kawasaki et al., Cell, 2020, PMID: 33159857) and *ASCL1* (Augustyn et al., Proc Natl Acad Sci U S A,

2014, PMID: 25267614) was observed, which highly supported their neuroendocrine origins (Extended Data Fig. 3a). Additionally, we showed relative low expression of the above marker genes in the PDAC organoid lines, while higher expression of many PDAC marker genes was confirmed. These results strongly helped to provide a credible identification of distinct transcriptome profiles among ACC, NEN and PDAC organoid lines and has been added to the Extended Data Fig. 3a in the revised manuscript.

Point 6: *The authors showed the association between chromatin accessibility peak and drug sensitivity. This data is interesting, but it is uncertain if these associations are specific or a random effect. Which gene was associated with each peak shown in Fig5h, i?*

NCOR gene regulates histone deacetylation other than JAK/STAT pathway. The authors should check if the Go6976 sensitive PDAC organoids have phosphor STAT3 upregulation. Similarly, they need to check the deacetylation status of MSH2 in Panobinostat sensitive PDAC organoids.

We thank Reviewer #1 for pointing out these valuable questions and helping us to improve our study. To address Reviewer's concern, we have carefully performed analysis on our data and validated the association between chromatin accessibility peak and drug sensitivity.

- 1) As suggested by the Reviewer #1, we tried to further investigate the cases in Fig. 5h, I, while we found that both the peak at chr1:179797737-179798364 and the peak at chr14:91694414-91695079 were not included in the peak-gene linkages of our PDPCOs, which might be due to the strict dual requirement on correlation and p-value for identifying significant peak-gene links. We then explored the regulatory function of these two peaks with public database. Among the four commonly used dataset about enhancer-gene links generated from normal tissues and tumor cell lines, five and six genes were showed associated with the peak at chr1:179797737-179798364 and the peak at chr14:91694414-91695079, respectively (Supplementary table 2), while only the correlation between RPS6KA5 gene and the peak at chr14:91694414-91695079 exhibited a p-value of 0.04199 in our PDPCOs with a low correlation of 0.3191065. Previous study has suggested that RPS6KA5 gene might have interactions with HDAC activity (Botia et al., PLoS One, 2012, PMID: 23110077) possibly supporting the relationship we observed between the peak at chr14:91694414-91695079 and the sensitivity of Domatinostat, a HDAC inhibitor. The above results indicated that the present understanding of enhancer-peak linkage was limited to study of conventional tumor cell line and normal tissues. Since huge difference existed between normal tissues/cell lines and tumor samples, further investigation into tumor samples, especially pancreatic cancer was urgently needed, where our PDPCOs has provided valuable resource.
- 2) Following the advice, we have carefully checked the phosphorylation of STAT3 among the 10 organoid lines in Extended Fig. 5g. To our surprise, notably higher phosphorylation of STAT3 was observed in the peak low group compared with peak high group (Referee Fig. 3a), which was different from our previous thoughts. However, a most recently published study revealed that the common function of NCOR2 was repressing gene expression and could work as a negative regulator of JAK/STAT3 signal pathway in estrogen receptor positive breast cancer (Tsoi et al., Cancers (Basel), 2021, PMID: 33806019). According to the above results, we inferred that the organoid lines in the peak low group might be free from the negative regulation of NCOR2 on JAK/STAT3 signal pathway, which resulted in higher phosphorylation of STAT3 and consequently resistant to the upstream inhibition (JAK inhibitor Go6976). In the peak high group, the JAK/STAT3 signal pathway in these organoid lines was suppressed by the high expression of NCOR2 leading to relatively low

phosphorylation of STAT3, which indicated that the oncogenic JAK/STAT3 signal pathway could function at activation and made them sensitive to Go6976. Therefore, we have rephrased relevant results from “which is involved in constitutive activation of the oncogenic JAK/STAT3 pathway in anaplastic large cell lymphomas” to “which is involved in regulation of the oncogenic JAK/STAT3 pathway in anaplastic large cell lymphomas and estrogen receptor positive breast cancer” and added the new reference in the revised manuscript. We also tried to carefully check the acetylation status of MSH2 in the 10 organoid lines involved in Extended Data Fig. 5j. Firstly, we performed western blotting of MSH2, in which PDPCOs in peak high group exhibited markedly higher level compared with those in peak low group (Referee Fig. 3b). Then we further tried to test their acetylation status. However, after multiple repeated attempts with acetylated-lysine antibody (CST, catalog: #9441), we found it was indeed hard to achieve reliable data. To some extent, a higher level of MSH2 protein in peak high group helped to provide us with further validation of the specific correlation between the chromatin accessibility peak at chr2:47355361-47355625 and MSH2 gene.

In summary, we have performed analyses on our ATAC-Seq data in this study, which demonstrated that the potential association between chromatin accessibility peak and drug sensitivity was more likely to be specific rather than a random effect.

Point 7: How accurately the high throughput analysis was carried out. Z' factor?

Is there any known gene-drug correlation, such as TP53 mutation vs Nutlin-3.

AUC value/drug/line (Extended Data Fig5 is hardly visible) should be shown in supplementary table.

In Fig6g, they showed the consistent drug responses in xenografts. CAS-DAC-36 showed slow tumor growth, and it is difficult to judge if they were resistant to chemotherapy or did not grow in the mice. Perhaps, the authors needed to observe for a longer duration when tumors grow slowly.

We thank Reviewer #1 for raising these important comments. To address Reviewer's concern, we have carefully performed analysis on our data and validated the gene-drug correlation both *in vitro* and *in vivo*.

1) The high throughput analysis was mainly conducted in the establishment of peak-to-gene links and peak-to-drug links where Z score method was routinely used. The z-score was defined as: $z = \frac{x - \bar{x}}{S}$, where \bar{x} was the mean of sample, S was the standard deviation of sample. For ATAC-seq peak, we calculated the z-score with openness across samples, while in gene expression, we used FPKM. Moreover, the drug AUC also was normalized as z-score.

2) In the drug screening of our PDPCOs, we indeed have tested many promising target inhibitors

including Nutlin-3 for another research. Although comparison of AUC between *TP53* wild type and *TP53* mutation organoid lines was not significant, which might be due to the limited size of *TP53* wild type cohort or some influence from other signaling pathway, a notable trend that *TP53* wild type organoid lines exhibited higher sensitivity to Nutlin-3 could be observed as expected (Referee Fig. 4a, b). Another well-known gene-drug correlation lied in *KRAS G12C* mutation VS AMG510 (Moore et al., Cancer Cell, 2021, PMID: 34375610). According to genomic analysis, we found that CAS-NEN-4 carried *KRAS G12C* mutation. Addition of such special organoid line into the screening of AMG510 (target therapy of *KRAS G12C* mutation recently approved by FDA (Blair et al., Drugs, 2021, PMID: 34357500)) revealed that organoid lines with *KRAS G12C* mutation showed extremely high sensitivity to AMG510 compared with those without *KRAS G12C* mutation (Referee Fig. 4c). According to the above results, we provided positive cases for known gene-drug correlation, which helpfully indicated that our PDPCOs could work as a credible high throughput drug screening platform. In this study, we mainly focused on the analysis of the selected epigenetic chemicals and chemotherapy, while detailed study of those promising target inhibitors were considering for another manuscript.

- 3) As suggested, we have supplied a detailed table (Supplementary Table 8) for the normalized AUC of drug screening in 39 exocrine PDPCOs with 59 chemical and 5 chemotherapeutic drugs in the revised manuscript.
- 4) To address the concerns about slow tumor growth in xenografts of CAS-DAC-36, we followed the Reviewer #1's suggestion to observe for a longer duration using ODX-36 generated with the same method in the manuscript. After prolonged 4 weeks of chemotherapy treatment, the vehicle group showed a notable increase in tumor volume for about 2.5 folds (Fig. 6g). Moreover, further analysis indicated that ODX-36 exhibited significant resistant to all of PTX, GEM and 5-FU with no statistical difference in tumor volume compared to Vehicle group (Fig. 6g and Extended Data Fig. 7f). These results provided solid evidence that ODX-36 was resistant to chemotherapy of PTX, GEM and 5-FU, which mirrored organoid sensitivity and patient clinical response, and we have supplied the new results to replace the original ones in the Fig. 6g and Extended Data Fig. 7f of revised manuscript.

In summary, we have shared some new data from *in vitro* and *in vivo* experiments to validate the gene-drug correlation and confirm the accuracy of our organoid sensitivity.

Reviewer #2

We find the paper “Integrated profiling of human pancreatic cancer organoids reveals chromatin accessibility to drug sensitivity networks” could be an important contribution to the field both as a great resource for the pancreatic cancer community and as an source of information for new transcription factor importance and mutations and their importance to different ATAC-seq peaks and their linkage to genes.

Major 1: HOMER motif score and motif enrichments. Figure 4 B/C shows motif enrichments calculated using rank TF lists as described in the methods. We wonder why not display HOMER enrichment from each cluster while using a background list of all peaks detected. This way we will have strong TF enrichments in each cluster over the genome and not a list of TFs that are enriched between clusters.

We thank Reviewer #2 for pointing out this important concern. In this study, we tried to unveil the specific

enriched transcription factor (TF) regulons in each cluster, in which RNA-seq data (generating TF regulon sets) and ATAC-seq data (generating enriched TF) were integrated as shown in Fig. 4a-c to help ensure a credible result. To generate enriched TF with ATAC-seq data, we firstly performed HOMER motif enrichment, in which we originally used random chromatin regions as the background. Following suggestions of the Reviewer #2, we repeated motif enrichment results with all peaks detected as the background, and comparison of motif enrichment p-value between these two settings showed that significantly high similarity could be observed among the four PDAC subtypes (Referee Fig. 5a and Referee table 3). For the part of exocrine and neuroendocrine organoid lines, we also found great similarity in the neuroendocrine group, while extremely low correlation was observed in exocrine group (Referee Fig. 5b and Referee table 3), which might be due to the overwhelming majority of specifically enriched peaks in exocrine group. We proposed that when using all peaks detected as background, significantly higher number of specifically enriched peaks in one group would absolutely impair its analysis of motif enrichment. Therefore, we preferred to using random chromatin regions as the

background. Secondly, we identified cluster specific enriched TF by the motif enrichment p-value. In addition to conventional threshold of p-value <0.05, we tried to employ a rank method (Kawasaki et al., Cell, 2020, PMID: 33159857) in this study to better elucidate the heterogeneity between clusters. As described in the method part, based on motif enrichment p-value, the rank method compared the enrichment p-value of one motif between clusters in detail and would allow the enrollment of relatively enriched TF whose p-value might be ≥ 0.05 . After identification of subtype-specific enriched TFs, TF regulon sets generated by RNA-seq data were further incorporated to finally reveal specific enriched TF regulons in each cluster. The whole analysis protocol strongly ensured the reliability of our study results, and a particular combination on inclusion criteria of enriched TF by ATAC-seq could widely help to facilitate more comprehensive understanding of heterogeneity in pancreatic cancer.

Major 2: We find the text description of Figure4 B/C confusing, while the data seen in Fig4 B/C seems to come from HOMER TF enrichment ($S_{i,j}$ in the methods) it is not described in the text and we find it confusing with the RNA/ATAC score ($SS_{i,j}$) in methods.

We Thank Reviewer #2 for the valuable comments and we are sorry for the confusing S_i^j and $SS_{i,j}$ methods. For ATAC-seq data, we used S_i^j to identify specific motif i enriched in given subtype j . Based on S_i^j , $SS_{i,j}$ was further calculated to reveal enriched TF regulon in specific subtype with both ATAC-seq data (generating S_i^j) and RNA-seq data (generating TF regulon sets). The combination of epigenomics and transcriptomics could significantly ensure the reliability of results we presented. In Fig. 4b, c, the enriched TF regulons of exocrine and neuroendocrine PDPCOs was shown in b, and the enriched TF regulons among the four PDAC transcriptome subtypes was shown in c.

Major 3: In their drug screen analysis, the authors use an approach that correlates cell drug sensitivity and ATAC-seq peaks. This approach limits the author resolution in ATAC-seq space and leads them to two classes drug sensitive/non-sensitive (as seen in Figure5G). This results worries us that most of the changes seen in the results is due to cell death/apoptosis and cell stress in response to the drug. We think there is a place here for a more cluster based analysis of only the ATAC-seq peaks, this analysis will give more emphasis to changes in ATAC-seq and will allow TF enrichment analysis, following this the authors can test which drugs are more correlated with the peaks change in each cluster.

We extremely appreciate Reviewer #2 for providing these important concerns. Actually, we do have tried to perform clustering with only ATAC-seq peak and we would like to share some results with Reviewer #2. The NMF clustering for the ATAC-seq peak suggested that $k=2$, $k=8$ and $k=4$ were the top three classifications (Referee Fig. 6a). Considering that $k=2$ provided limited insights on revealing pancreatic cancer heterogeneity and $k=8$ divided the samples too dispersedly, we have chosen $k=4$ as the classification for the ATAC-seq peak (Referee Fig. 6b). Combined with motif enrichment from subtype-specific peak and corresponding TF expression, the 10 most enriched TFs were showed in the four classes, in which we surprisingly found that a large amount of the enriched TFs overlapped among the different classes (Referee Fig. 6c). Since all of the used organoid lines were generated from the same pathological type-pancreatic cancer, high similarity of enriched TFs between clusters such as FOS (Baron et al., Cell Syst, 2020, PMID: 32910905), KLF5 (He et al., Gastroenterology, 2018, PMID: 29248441) and EHF (Liu et al., J Exp Med, 2019, PMID: 30733283), which might play critical roles in pancreatic lineage

and tumorigenesis, could be possible. Moreover, according to the difference between clusters, we still found some specific correlations between TFs and drugs. In the class4, we identified the enrichment of MYC, which has been reported to regulate downstream effectors by promoting expression of HDAC5 (Xue et al., Br J Cancer, 2019, PMID: 31690832). As expected, comparison of drug sensitivity to AR-42, a novel class of HDAC inhibitor revealed significantly higher sensitivity in class4 rather than the other three classes. Another case lied in the correlation between AP-1 complex (FOS/FOSL1/JUN) and PKC, which was previously studied that PKC was crucial for the activation of AP-1 complex (Sokolova et al., Gut, 2013, PMID: 22442164). In the drug screening of Go6976, a PKC inhibitor, we showed that the AP-1 complex

Referee Fig. 6 | ATAC-seq peak based classification of the PDPCOs and motif enrichment. a, Detailed data from unsupervised NMF clustering in the 40 exocrine PDPCOs (39 CAS-DACs and 1 CAS-IPMN). Solutions were shown for classes k=2, k=4 and k=8. **b,** Heatmap showing class-specific peaks when choosing k=4 as the classification for ATAC-seq peak. **c,** Scatter plot showing the motif enrichment and expression of corresponding TF in different classes when choosing k=4 as the classification for ATAC-seq peak. **d,** Comparison of the AUC of AR-42 between class1+class2+class3 and class4 in the 39 exocrine PDPCOs (38 CAS-DACs and 1 CAS-IPMN) with drug test data. **e,** Comparison of the AUC of Go6976 between class1+class2 and class3+class4 in the 39 exocrine PDPCOs (38 CAS-DACs and 1 CAS-IPMN) with drug test data.

enriched class1 and class2 were more sensitive compared with class3 and class4. The above association between ATAC-seq cluster and drug sensitivity provided some further evidence for the credibility of our high throughput drug screening. However, considering the relative high overlap of enriched TFs between clusters and a few target inhibitors for the lineage and tumorigenesis associated TFs, we preferred to choosing a novel resolution in our manuscript. Since regulatory elements were well known to function in cellular physiological processes and many of them could even act as a live-or-die checkpoint, we have tried to focus on investigation of specific correlation between recurrent ATAC-seq peak and drug sensitivity. The identified peak-drug relationship could not only help to shed light on potential mechanisms of underlying gene regulatory networks, but also provide a new direction for developing target therapy with ATAC-seq peak as the special biomarker. In this study, the feasibility of such resolution has been proven by several well demonstrated cases of peak-gene links including both chemotherapy and chemicals, and further investigation was needed to help elucidate more practical networks.

Minor 1: *WGS comparison between organoids and tissue: The authors choose to show the similarities in genomic scale but both class specification and gene level analysis (Extended figure 2B and 2C) show differences between the organoids to the tissue. It will be useful to have a more accurate analysis showing the similarities and changes between the tissues to organoids.*

We are grateful to Reviewer #2 for this helpful advice to improve our study. To more accurately assess the similarity between our PDPCOs and paired tumor tissues, we examined the purity of PDPCOs (WGS) and tissues (WES), and further compared their concordance in mutations among exon regions. As expected, the estimated cellularity of these tissues was relatively high at around 40%, since we have selected samples with enough tumor cells based on histological examination (Extended Data Fig. 2d). Besides, extremely high purity was observed in our PDPCOs, which has been reported one of advantages for organoid model (Extended Data Fig. 2d). Comparison of detailed mutations among exon regions between organoids and matched tissues showed that majority of the 10 pairs achieved favorable concordance of 50.00%-60.00%, and the organoid line derived from ACC which was characterized by abundant tumor cell exhibited the highest concordance of 74.60% (Extended Data Fig. 2d). Moreover, we found that most of mutations in exon regions of tumor tissues could be similarly detected in their corresponding organoid lines, in which 6 pairs reached to more than 70% (Extended Data Fig. 2d, e). The above results, which has been added to Extended Data Fig. 2 of the revised manuscript, provided a more accurate analysis of similarities and changes between the tissues and organoids, and helped to demonstrate that our PDPCOs could maintain the genomic features of original tumors.

Minor 2: *The authors claim 527,272 new non coding mutations and supply a detailed table where one can generate all the info but some information about the mutations and how many of them are only found in one sample and how many are shared is missing.*

We thank Reviewer #2 for the constructive suggestion. We have supplied the detailed information about shared noncoding mutation in Supplementary table 2. In detail, we identified 527,272 noncoding mutations in total, among which 5,262 noncoding mutations were found in at least two organoid lines and the remaining 522,010 noncoding mutations were only found in one sample.

Minor 3: *The authors define 4 sub class of pancreatic cancer claiming that 2 of the subtypes are new, the*

authors fail to compare their new subtypes to many of the proposed subtype definitions that are found in the literature, for a nice summary of the different classifications (30419209)

We thank Reviewer #2 for pointing out this valuable concern. To clarify the association between our transcriptomic subtypes and the reported classifications, we clustered our PDPCO based on signature genes from previous studies (Referee table 1). Since organoid model was characterized by high cell purity, we specially used “Squamous subtype” and “Progenitor subtype” in Bailey et al. classification (Bailey et al., Nature, 2016, PMID: 26909576), in which “ADEX subtype” and “immunogenic subtype” likely represented gene expression from non-neoplastic cells (Aguirre et al., Gastroenterology, 2018, PMID: 30419209), as well as “Classical subtype” and “Basal-like subtype” in Puleo et al. higher cellularity tumor classification (Puleo et al., Gastroenterology, 2018, PMID: 30165049). Another two classifications from Moffitt et al. (Moffitt et al, Nat Genet, 2015, PMID: 26343385) and Collisson et al. (Collisson et al., Nat Med, 2011, PMID: 21460848) were also examined with our samples. According to the results (Referee Fig. 7), we learned that the “Classical-like subtype” and “Classical-Progenitor subtype” in our study were highly consistent with previous “Classical/Progenitor subtype”, and “Basal-like subtype” were closely related to the reported “Basal-like subtype”. For the newly defined “Glycomet subtype”, it was more inclined to previous “Basal-like subtype” in the classifications of Bailey et al., Moffitt et al. and Puleo et al., while it showed multiple expression signatures in Collisson et al. classification. The above results strongly indicated a close relationship between our RNA subtypes and previous classification, and highly

supported the potential existence of a hybrid subtype identified by our study, which was characterized by activated glycometabolic pathway.

Minor 4: Extended Data Fig. 4 panel b and c are in reverse order in the legend

Page 7 Line 21, Sends reader to Fig4a while meant 4b

We have corrected it in the revised manuscript.

Reviewer #3

The authors establish patient-derived pancreatic cancer organoid lines and performed an integrative analysis of paired chromatin accessibility, transcriptomic, genomic profiles and high-throughput drug screening. Overall, the study provides a useful organoid with some novel findings that demonstrate the utility of organoids in predicting therapeutic outcomes in pancreatic cancer. However, the rationale for some of the approaches is not clear and impacts the ability to assess the significance of the relationships highlighted and conclusions drawn from them.

Major 1: Tissue microenvironment can contribute to a large confounding effect in cancer epigenomic studies due to different cell types have distinct epigenetic signatures. The authors have indicated that analyzing ATAC-seq profiles in primary pancreatic tumor tissue can be challenging due to presence of excessive stroma. The authors should report on the (microenvironment) composition of the patient-derived organoid lines generated compared to the primary tumors, and state what measures were taken place to account for any confounding effects (if present).

We are grateful to Reviewer #3 for raising this important concern. As mentioned by Reviewer #3, low cellularity of tumor samples could exert large confounding effects on the investigation of cancer epigenomics due to the existence of many other cell types, which might be the reason for the lack of study on pancreatic cancer epigenomics. To solve this problem, we used organoid model generated from primary pancreatic cancer samples. It has been reported that organoid model could significantly enrich epithelial cells and achieve high purity of tumor cell. With flow cytometry analysis of the live single cells, we showed that tumor tissue was a complicated mixture of epithelial cell (Epcam+), while blood cell (CD45+) and other cell types such as stromal cell, while the organoid generated from tumor tissue exhibited a significant enrichment of epithelial cell to 96.4% (Referee Fig. 8a). Furthermore, according to the genomic analysis of 10 PDPCOs and matched tissues, we demonstrated that our PDPCOs could reach to a purity of almost 100%, which were independent of the cellularity of tumor tissues (Referee Fig. 8b). The above results suggested that our PDPCOs harbored extremely high purity of tumor cell

compared to primary tumors. With histological examination and genomic analysis demonstrating great similarity between tissues and matched organoids, ATAC-seq profiles of our PDPCOs would hopefully promote more comprehensive understanding of pancreatic cancer epigenomics, and we are also looking forward to further investigation into it with co-culture model of organoids and other components.

Major 2: *The authors should clarify how many tests were performed to identify subtype-specific TF regulons (for $p_{(i,j)}^1$ and for $p_{(i,j)}^2$). For a large number of tests, the authors should use FDR (or similar) adjusted p-values.*

We appreciate Reviewer #3 for this helpful suggestion to improve our study. In this study, we inferred 377 TF regulons from gene expression profiles, 283 of which were available for motif enrichment analysis with HOMER and statistical analysis to identify subtype-specific TF regulons. Following Reviewer's suggestion, we used Benjamini-Hochberg procedure (FDR) to adjust the original p-value results (Referee table 4) and indeed found some differences in comparison of enriched TF regulons between original results and adjusted results. In the adjusted results, several TF regulons were showed with a q-value ≥ 0.05 , while most of them were demonstrated to have limited influence on further understanding subtype-specific characteristics in our original analysis. It should be noted that, for classical-like subtype, the former YY-1 TF regulon were removed due to q-value ≥ 0.05 , while HNF4A and HNF4G TF regulons were still highly enriched, which provided credible attribution of this subtype-conventional classical classification (Puleo et al., Gastroenterology, 2018, PMID: 30165049). Importantly, for the glycomet subtype, the originally identified TF regulons were found with borderline significance in the adjusted analysis and we therefore made further exploration to unveil its potential biological characteristics. After careful investigation, many TFs that were reported correlated with tumor invasion and metastasis of other solid cancers were significantly identified in the glycomet subtype, which strongly help to interpret its poor prognosis. The above results proposed that adjusted p-value should be used instead of p-value for a large number of tests, and we have rephrased the results of Supplementary table 4 in the revised manuscript.

Major 3: *The authors calculate correlations over all possible interactions between ATAC-seq peak accessibility and gene expression within 500 kb of the peaks (after filtering for low variance) to identify putative links. The authors should state the total number of interactions tested and use FDR (or similar) adjusted p-values.*

We thank Reviewer #3 for this constructive advice. We have applied Benjamini-Hochberg procedure (FDR) (Corces et al., Science, 2018, PMID: 30361341) to adjust the p-values of putative links between ATAC-seq peak accessibility and gene expression within 500 kb of the peaks, in which 2,257 interactions were tested consistent with original analysis. Although there were slight differences compared with former statistical results, their adjusted p-value all remained less than 0.05 (Referee table 4), which was likely caused by the former analysis procedure that we firstly filtered the peak-to-gene linkage by > 0.5 or < -0.5 Pearson correlation coefficient, and then calculated the p-value of the selected linkages for further analysis. According to the above results, we strongly agree with Reviewer #3 that it is more reasonable to use adjusted p-value for a large number of tests and have replaced the p-value with FDR adjusted p-value in Supplementary table 4 in the revised manuscript, while in this study an additional constrain on Pearson correlation coefficient helped in keeping the final results of peak-to-gene linkage unchanged.

Major 4: *The authors calculate correlations over possible interactions between ATAC-seq peak accessibility and drug sensitivity to identify putative links. The authors should state the total number of interactions tested and use FDR (or similar) adjusted p-values.*

We thank Reviewer #3 for this constructive suggestion. Following this advice, we have calculated the adjusted p-values of interactions between ATAC-seq peak accessibility and drug sensitivity by Benjamini-Hochberg procedure (FDR) (Corces et al., Science, 2018, PMID: 30361341). Consistent with former analysis, a total of 15,397 peak-to-drug links were tested. Although slight variations could be observed, the adjusted p-value all remained less than 0.05 (Referee table 4), which was similar to situation in Major 3. We proposed the similar reasons for this case that we screened the peak-to-drug links by > 0.5 or < -0.5 Pearson correlation coefficient, and then calculated the p-value of the selected linkages. Based on these results, we believe that adjusted p-value should be routinely used for large amounts of tests and therefore have corrected p-value used in Supplementary table 4 in the revised manuscript, while the detailed threshold on Pearson correlation coefficient similarly helped in stabilizing the results of peak-to-drug links.

Major 5: *In Figure 5j, 10 representative samples were chosen to show case differential accessibility at a putative enhancer locus. Could the authors clarify how these samples were chosen? Were these the samples that had the most extreme differences? The authors should repeat the boxplots Figure 5k,l,m over all samples (if necessary stratify the ATAC-peak into tertiles – high, medium, low). Could the authors also clarify/ repeat this for Extended Data Figures 5g-q.*

We are grateful to Reviewer #3 for raising these important questions. To present notable visual discrepancy between the peak high group and the peak low group, we chose to show the samples with the most extreme differences, while vast majority of these cases also exhibited significant differences over all samples. As suggested by Reviewer #3, we divided all samples into three groups based on the ATAC-seq peak openness to compare their corresponding gene expression and drug sensitivity. For the peak at chr10:121441617-121442244, we observed various gene expression of *BAG3* gene between peak high group and peak low group with borderline significance among all samples (Referee Fig. 9a), while the corresponding drug 5-FU and PTX exhibited extreme difference (Referee Fig. 9b). Besides, we also checked the results of the other three cases over all samples. The significant difference on *NCOR2* gene expression was observed between the high group and the low group of the peak at chr12:125033345-125034019 (Referee Fig. 9c), and their drug sensitivity to Go6976 highly varied (Referee Fig. 9d). Moreover, we found similar results in the analysis of peak at chr6:139728666-139729424, *CITED2* gene expression and OXA sensitivity were found markedly different between the peak high group and the peak low group among all samples (Referee Fig. 9e, f). For the peak at chr2:47355361-47355625, higher expression of *MSH2* gene was confirmed in the peak high group (Referee Fig. 9g), and significant difference on Panobinostat, Abexinostat and Quisinostat between the peak high group and the peak low group could be observed (Referee Fig. 9h). According to the above results, we proposed that the significant difference on gene expression and drug sensitivity among 10 representative samples could be repeated over all samples.

Referee Fig. 9 | Comparison of gene expression and drug sensitivity based on tertiles of ATAC-seq peak openness over all samples. Stratifying the openness of ATAC-peak at chr10:121441617-121442244 into tertiles over all samples (low, medium and high) to compare the expression of BAG3 gene in **a** and the drug sensitivity of 5-FU and PTX in **b**. Stratifying the openness of ATAC-peak at chr12:125033345-125034019 into tertiles over all samples (low, medium and high) to compare the expression of NCOR2 gene in **c**, the drug sensitivity of Go6976 in **d**. Stratifying the openness of ATAC-peak at chr6:139728666-139729424 into tertiles over all samples (low, medium and high) to compare the expression of CITED2 gene in **e**, the drug sensitivity of OXA in **f**. Stratifying the openness of ATAC-peak at chr2:47355361-47355625 into tertiles over all samples (low, medium and high) to compare the expression of MSH2 gene in **g**, the drug sensitivity of Panobinostat, Abexinostat and Quisinostat in **h**.

Major 6: The authors have proposed specific examples of putative chromatin accessibility peaks as predictive biomarkers of drug response in the PDCPOs. Can the authors comment on whether these epi(genomic) alterations presented intrinsic resistance/ sensitivity or could they be inducing resistance to the therapy. And specifically discuss their application in novel treatment strategies?

We thank Reviewer #3 for pointing out these questions. We have specifically discuss the potential application of putative chromatin accessibility peaks in guiding treatment strategies in the revised manuscript. In this study, all of the generated organoids involved in chemicals screening were cultivated

under the same condition without special pre-treatment, which strongly proposed that the difference on chromatin openness between organoid lines presented intrinsic epigenomic heterogeneity. However, since epigenomic alterations were known potential targets that could be pharmacologically reversible, it was also possible to induce sensitivity or resistance of some drugs by manipulating corresponding chromatin accessibility. Compared with traditional genomics and transcriptomics, which have been widely investigated to identify biomarkers for effective treatment, such chromatin accessibility has brought us a completely novel approach for developing cancer treatment. Based on comprehensive understanding of gene regulatory network in pancreatic cancer, we would be able to uncover specific regulatory elements responsible for treatment sensitivity or resistance, which promisingly help to provide a greater chance for exploring more therapeutic targets and promoting research on effective chemicals.

Major 7: The authors have compiled and performed drug screening on a large library of chemicals targeting epigenetic-related signaling pathways, with a decent number (59) showing inhibition in 1 or more PDCPOs. The authors should provide a figure for the resistance/ sensitivity for these agents in the PDCPOs similar to Figure 6a.

Since the chemical library we used has not been systematically investigated in human tumor organoids, especially in pancreatic cancer organoids. We determined to category the sensitivity of our PDPCOs to these chemicals referring to the method for chemotherapy. According to drug AUC, we equally stratified our PDPCOs in each chemical

into three sensitivity groups (high as resistant, medium as intermediate, low as sensitive). The chemical sensitivity heatmap notably showed that different chemicals had various effects on organoid lines, and each PDPCOs exhibited diverse sensitivity to different chemicals (Referee Fig. 10). Although such division of sensitivity was rather a relative result, it helped to provide an overview for the sensitivity of the 59 chemicals in our PDPCOs and would promote further investigation into the potential heterogeneity in pancreatic cancer.

Major 8: Do the authors have any data on the consequence of any epigenetic-related therapies on the epigenome of the PDCPOs? For instance, ATAC-seq of the PDCPOs before and after treatment for epigenetic-therapies demonstrating strong inhibition.

We appreciate Reviewer #3 for providing this constructive suggestion. To further investigate the correlation between drugs and corresponding peaks, we collected some ATAC-seq data from PDPCOs

after drug treatment. We mainly focused on the association between chr10: 121441617-121442244 and 5-FU. After treatment of 5-FU (10uM) for 3 days, we performed ATAC-seq on the 10 samples involved in Fig. 5j. By comparison of the corresponding peak openness, we found that, in both peak high and peak low groups, significant higher openness could be observed in the groups receiving drug treatment (Referee Fig. 11), which supported their specific correlation generated from high throughput drug screening in the manuscript that the peak at chr10: 121441617-121442244 could negatively predict the sensitivity of 5-FU. Moreover, in the treatment naïve samples, we proposed that *BAG3* gene worked as a connector for the correlation between peak and drug. Although the strong association between peak and drug could also be observed after drug treatment, the detailed mechanism of *BAG3* gene in the post-treatment correlation need to be further verified.

Major 9: Were post treatment imaging results available for more than 3 patients: CAS-DAC-24/22/20? If so, the authors should provide a graphical summary of the correlation between the imaging results and organoid sensitivity for all patients with available paired data treated with the same (or similar mode-of-action) chemotherapy agents as they have done for these 3 examples.

We are grateful to Reviewer #3 for pointing out this important question and help us to improve our study. The department of Hepatobiliary Pancreatic Surgery of Changhai Hospital is one of the most popular centers for pancreatectomy in China, in which patients were from all over the country. Since majority of patients lived far from Shanghai, they would like to receive routine laboratory and image examinations in local hospital and were followed up by telephone for convenience. After carefully checking our prospective database for image examinations, we found another three patients (CAS-DAC-5, CAS-DAC-21, CAS-DAC-31) with available paired data. CAS-DAC-31 was assigned to sensitive group due to the use of organoid sensitive S1 in paired patient. CAS-DAC-5 was in intermediate group, the S1 patient used exhibited moderately sensitive in matched organoid. For CAS-DAC-21, the corresponding patient used GEM and S1 for adjuvant chemotherapy, while both of them showed limited effect on organoid growth. According to the six patients (CAS-DAC-5, CAS-DAC-20, CAS-DAC-21, CAS-DAC-22, CAS-DAC-24 and CAS-DAC-31) with available paired data, we analyzed the correlation between imaging results and organoid sensitivity within the same chemotherapy agent. S1 was used in CAS-DAC-5 (intermediate), CAS-DAC-20 (intermediate), CAS-DAC-21 (resistant), CAS-DAC-22 (resistant) and CAS-DAC-31 (sensitive), while GEM in CAS-DAC-24 (sensitive) and CAS-DAC-21 (resistant). Comparison of image examinations significantly demonstrated a strong consistency between organoid

sensitivity and recurrence status under the same chemotherapy agent, in which “sensitive” showed less recurrence than “intermediate”, while “resistant” exhibited the most progression (Extended Data Fig. 6d, e). The above results further supported that the high consistency in chemosensitivity between the organoids and paired patients.

Minor 1: What was the rationale to use $\log_2(\text{copy number})/2 > 0.1$ as a threshold for gains/ amplifications in the Extended Data Figure 2. Would $\log_2(\text{copy number}/2) > 0.1$ be a better threshold?

Reviewer #3 is correct. We have rewritten this point in the revised manuscript.

Minor 2: The authors indicate that the somatic mutational profiles of the PDPCOs were similar to that observed in the TCGA pancreatic adenocarcinoma (PAAD) dataset. Could the authors include evidence for this conclusion?

We thank Reviewer #3 for this important comment. To better elucidate the similarity of somatic mutational profiles between our PDPCOs and the TCGA PAAD dataset, we provided a detailed comparison in Referee Fig. 12. Among the four driver genes of pancreatic cancer (KRAS, TP53, SMAD4 and CDKN2A), our PDPCOs exhibited the exact same mutated tendency to TCGA PAAD dataset, in which slightly higher mutation frequencies could be observed in our PDPCOs likely due to the increased cellularity (Tiriach et al., Cancer Discov, 2018, PMID: 29853643; Chan-Seng-Yue et al., Nat Genet, 2020, PMID: 31932696). For the other top mutated genes, our PDPCOs showed similar results compared with TCGA PAAD dataset (Referee Fig. 12). According to the above results, we proposed that the somatic mutational profiles of our PDPCOs were similar to that observed in the TCGA PAAD dataset.

Minor 3: Line 1-2 on page 6 is unclear. Can the authors consider rephrasing?

We thank Reviewer #3 for raising this helpful comment. We are sorry for the confusing expression and have corrected it in the revised manuscript.

Minor 4: Extended Data Figure 3a details multiple solutions for determining the number of clusters for NMF of the transcriptomic profiles for the exocrine group. The authors should clarify why $k=4$ was chosen over the other solutions? Also check the legend for $k=3,4$ in the figure.

We are grateful to Reviewer #3 for pointing out these concerns.

1) As shown in Extended Data Fig. 3, $k=2$ and $k=4$ were the best two classifications for the

transcriptomic profiles for the exocrine group. We firstly investigated the characteristics of the k=2 classification (Referee Fig. 13a). As expected, the two clusters were found closely related to the previously reported classical subtype and basal-like subtype, respectively (Referee Fig. 13b), which indicated high consistency between our PDPCOs and primary pancreatic cancer. However, we learned from recent studies that more detailed classifications in pancreatic cancer including subdivision of conventional classical and basal-like subtypes have been suggested to promote better understanding of tumor heterogeneity and provide more accurate prognostic prediction (Chan-Seng-Yue et al., Nat Genet, 2020, PMID: 31932696). Since organoid could well recapitulate the primary tumor and feature the high cellularity, we tried to further explore the transcriptomic classification of pancreatic cancer with our PDPCOs and then chose k=4 for the classification of exocrine group. According to comparison with previous subtypes and signal pathway enrichment analysis, we also proposed a subdivision of classical subtype into classical-progenitor and classical-like, and successfully identified a novel transcriptomic subtype-glycomet subtype. Moreover, further survival analysis revealed that only the classical-progenitor subtype achieved the most favorable outcomes, while basal-like subtype, glycomet subtype and the other classical subtype named classical-like subtype had similar poor prognosis.

2) We are sorry for the mistake about the legend for k=3, 4 in the Extended Data Fig. 3 and have

corrected it in the revised manuscript.

Minor 5: In Figure 3b/ Extended Data Figure 3b, the heatmap columns are indicated by samples, however, the legend states otherwise.

We apologize to Reviewer #3 for this error and have corrected it in the revised manuscript.

Minor 6: *Figure 4b is not mentioned in the text.*

We thank Reviewer #3 for the kind reminding and have added it in the revised manuscript.

Minor 7: *In line 27, page 8 the authors state that “95% of genes were regulated by multiple peaks (≤ 6) in the distal ATAC-seq peak to-gene links”. However, this conclusion is incorrect since most of these genes were regulated by only 1 peak.*

Reviewer #3 is correct. We have rewritten this point by “95% of genes were regulated by no more than 6 different peaks in the distal ATAC-seq peak-to-gene links” in the revised manuscript.

Minor 8: *The authors should clarify what is the evidence to back the claim “ATAC-seq peak at chr6:139728666-139729424 predicted sensitivity to oxaliplatin” in line 23, page 6.*

We thank Reviewer #3 for pointing out this important concern. In the linkage analysis of ATAC-seq and RNA-seq, we found that the ATAC-seq peak at chr6:139728666-139729424 had a direct positive correlation with the expression of the CITED2 gene. Moreover, Previous study had demonstrated that CITED2 was overexpressed oxaliplatin (a third-generation platinum drug) resistant cancer cell lines compared to oxaliplatin sensitive cancer cell lines, while knock down of CITED2 could enhance the cytotoxicity of platinum (Yanagie et al., Biomed Pharmacoth, 2009, PMID: 18571892). According to above results, we proposed that ATAC-seq peak at chr6:139728666-139729424 would have potential utility in predicting sensitivity to platinum.

Minor 9: *Figure 6a legend is misleading in how PDCPOs were divided into 3 categories for chemo sensitivity/ resistance. The figure suggests this is done by tertiles, while the text/ figure annotation suggests that is done by AUC value thresholds of 33%,67%. Consider rephrasing.*

We have rephrased the category threshold from “33%, 67%” to “AUC lowest 1th-13th, 14th-26th, 27th-39th” in the revised manuscript.

Minor 10: *The authors demonstrate CAS-DAC-14, as an example with significant inter-agent variability within a single organoid. The authors should include a figure demonstrating this graphically for all organoids, and link each patient with the 3 groups defined for Figure 6c to ease the understanding of the group definitions. For instance, it is not immediately clear what patients were included in the resistance group – did these patients demonstrate resistance in the paired organoids for ALL chemotherapy agents?*

We have organized a graphical summary for better understanding of group definition and chemotherapy administration in 31 patients who underwent upfront radical surgery with subsequent adjuvant therapy. Significant inter-agent variability within a single organoid could be observed for most organoids, while several organoids exhibited same sensitivity to all five chemotherapy agents. For instance, CAS-DAC-15 and CAS-DAC-18 were sensitive to all five chemotherapy agents, whereas CAS-DAC-2, CAS-DAC-13 and CAS-DAC-35 showed resistant to all five chemotherapy agents. The graphical summary has been added to Extended Data Fig. 6a in the revised manuscript.

Minor 11: *The authors generated organoid derived xenografts (ODX) for 3 patients: CAS-DAC-18/20/36. The authors should clarify whether ODXs were attempted/ generated for other patients and why these 3 were chosen to be described in the manuscript (others had also shown similar resistance/sensitivity to chemotherapy agents such as CAS-DAC 22/24).*

We appreciate Reviewer #3 for raising these important questions to help improve our study. Actually, we have tested the generation success rate with the first 37 cases of PDAC organoids as well as one case of ASC and one case of IPMN organoids by subcutaneously injection into SCID mice, in which almost all of the PDPCOs developed xenografts within 4-16 weeks (97.4%, 38 of 39), except for one case (CAS-DAC-5, PDAC). Considering the favorable success rate and high cost of in vivo experiment, we performed the in vivo validation of organoid sensitivity with randomly chosen cases among the three sensitivity groups. As suggested by Reviewer, we further validated the organoid sensitivity of CAS-DAC-22 and CAS-DAC-24 with their corresponding xenograft, in which CAS-DAC-22 was resistant to 5-FU, while CAS-DAC-24 was sensitive to CAS-DAC-24. As expected, ODX-22 showed notably resistant to 5-FU with a reduction in tumor growth of only 17.04%. For the ODX-24, significant inhibition on tumor growth could be observed by GEM with the reduction as high as 75.58%. These results provided further evidence to confirm the accuracy of our PDPCO chemotherapy sensitivity and has been added to Extended Data Fig. 7 in the revised manuscript.

Minor 12: *What timepoint was used for measuring reduction in tumor growth in the ODXs post treatment. ODX-36 response looks similar at day 20 for all 3 agents shown, but the authors indicate a notable difference. Could the authors clarify this?*

We thank Reviewer #3 for pointing out this valuable concern. For all of the three ODXs, we have performed comparison of the reduction in tumor growth at the last measure time. Although the response to the three chemotherapy agents of ODX-36 looked similar at day 20, according to the statistical analysis, we found that GEM slightly reduced tumor growth in ODX-36 (32.60% reductions in tumor growth, $P=0.031$), while ODX-36 displayed significant resistance to the other two drugs (5-FU and PTX, $P=0.226$ and $P=0.132$), as demonstrated by 20.09% and 27.47% reductions in tumor growth (NOTE: we appreciated Reviewer #3 for the comments, since we realized that the original 8.66% and 20.15% were results of another timepoint during statistical analysis). To ease the understanding of comparison, the results of statistical analysis for all of the three ODXs have been added into the Fig. 6 in the revised manuscript, in which Fig.6g of ODX-36 has been updated with results from longer observation to minimize the effect of slow tumor growth showing 7.79% reductions in tumor growth for GEM ($P=0.566$), 24.51% for 5-FU ($P=0.183$) and 25.17% for PTX ($P=0.077$).

REVIEWER COMMENTS

Reviewer #1, expert in organoids (Remarks to the Author):

The authors addressed all points that I raised in the previous review.

Reviewer #2, expert in ATAC-seq analysis (Remarks to the Author):

The authors have addressed my concerns

Reviewer #3, expert in multi-omics and drug response (Remarks to the Author):

The authors have provided a revised manuscript describing the establishment of patient-derived pancreatic cancer organoid lines; and integrative analysis of paired chromatin accessibility, transcriptomic, genomic profiles and high-throughput drug screening. The authors responded to each of the specific comments raised by the reviewers and revised the manuscript accordingly. Further comments included below.

1. In response to the reviewer's comment for Major 3, the authors have indicated that they first filtered putative peak-expression links based on pearson correlation coefficient, and then calculated p-values on the remaining linkages and performed FDR adjustment. Could the authors clarify whether the p-values were also based on pearson correlation test for the remaining linkages? If this is the case, filtering using estimates from the statistical test prior to p-value calculation from the same statistical test is not standard practice, and defeats the purpose of multiple testing. The authors could consider a more stringent filtering on variance on the peaks and expression values separately.

2. Same concern is noted for Major 4 as for Major 3 (see above).

3. The authors have responded adequately to Major 5. The authors should clarify in the manuscript that the 10 representative examples used were picked to show the most extreme differences. Referee Fig 9 is very encouraging; the authors could consider including this in Supplementary.

4. The authors have indicated in the manuscript that the somatic mutational profiles of the PDCPOs were similar to that observed in the TCGA pancreatic adenocarcinoma (PAAD) dataset. To back this claim, the authors should add Referee Fig 12 to Supplementary.

My overall impression remains that the manuscript is a matter of interest since they provide a useful organoid resource with some novel findings that demonstrate the utility of organoids in predicting therapeutic outcomes in pancreatic cancer.

Point-by-point response to Reviewers' comments

Nov 6th, 2021

RE: NCOMMS-21-10013A

Integrated profiling of human pancreatic cancer organoids reveals chromatin accessibility to drug sensitivity networks

Summary statement. We are grateful to the Editor and Reviewers for their thoughtful comments which pointed to two key issues: 1) improvement of our filtering and statistical testing; 2) supplement of research data involved in referee figures. Based on these issues, we have worked diligently to address these issues. Briefly, we have supplied additional data in our manuscript and made the following changes:

- Improved our filtering and statistical testing procedure for identifying gene-peak linkage and peak-drug linkage.
- Supplied additional research data from referee figures about correlation between ATAC-seq peak and drug as well as comparison of somatic mutational profiles between PDPCOs and TCGA PAAD dataset.

Please see below for the detailed responses to Reviewer #3's comments

Reviewer #3

The authors have provided a revised manuscript describing the establishment of patient-derived pancreatic cancer organoid lines; and integrative analysis of paired chromatin accessibility, transcriptomic, genomic profiles and high-throughput drug screening. The authors responded to each of the specific comments raised by the reviewers and revised the manuscript accordingly. Further comments included below.

Point 1: *In response to the reviewer's comment for Major 3, the authors have indicated that they first filtered putative peak-expression links based on Pearson correlation coefficient, and then calculated p-values on the remaining linkages and performed FDR adjustment. Could the authors clarify whether the p-values were also based on Pearson correlation test for the remaining linkages? If this is the case, filtering using estimates from the statistical test prior to p-value calculation from the same statistical test is not standard practice, and defeats the purpose of multiple testing. The authors could consider a more stringent filtering on variance on the peaks and expression values separately.*

We are grateful to Reviewer #3 for the shrewd observation on this multiple testing problem and suggestions for solving it in a more rigorous way. We clarify that our adjusted p-values were based on Pearson correlation test for the remaining linkages. We agree that filtering using estimates from the statistical test prior to p-value calculation from the same statistical test is not standard practice. The estimates used for filtering should provide information on the statistical properties of hypothesis tests and should be independent of the p-value under the null hypothesis (Bourgon et al., Proc Natl Acad Sci U S A, 2010,

PMID: 20460310).

Following Reviewer #3's suggestions, we apply more stringent filtering and statistical tests to improve our research. In this study, there are 15,609 genes and 156,721 peaks used for gene-peak correlation analysis. We first filter all 2,446,258,089 gene-peak pairs by genome distance, which is a widely used strategy in ATAC-seq, eQTL and GWAS studies (Corces et al., Science, 2018, PMID: 30361341; Korthauer et al., Genome Biol, 2019, PMID: 31164141; Ignatiadis et al., Nat Methods, 2016, PMID: 27240256). As a result, 1,019,530 pairs that peak is within 500 kbp of the gene's TSS remain and are used to calculate Pearson correlation, p-values and FDR. To increase detection power, we then try to filter the gene-peak pairs by variance on the peaks and expression values by removing the bottom 10%, 15%, 20%, 25%, 30%, 35%, 40%, 45%, 50%, 55%, and 60% of genes and peaks with low variance. In the following table (Referee table 1), the number of identified gene-peak linkages and their overlap with the identified 2,257 linkages based on different cut-offs of FDR and absolute PCC>0.5 are showed, which suggest that stringent filtering on variance could also help to reduce the testing gene-peak pairs. Finally, to achieve a good balance between coverage and accuracy for identifying putative gene-peak links, we choose to remove the 25% genes and peaks with lowest variance (Corces et al., Science, 2018, PMID: 30361341) and modify FDR cut-off to 0.25 (Cao et al., Cell, 2021, PMID: 34534465) as previous studies suggested, which result in the same 2,257 gene-peak linkage set.

In summary, we follow reviewer's suggestions and clarify a more rigorous procedure for filtering and statistical tests in the method part of the revised manuscript. We now filter the gene-peak pairs by two independent co-variants (genome distance and variance on the peaks and expression values) and then perform p-value adjustment to improve our study.

Referee table 1. The number of identified gene-peak linkages and their overlap with the identified 2,257 linkages based on different cut-offs of FDR and absolute PCC>0.5 under removing the bottom 10%-60% of genes and peaks with low variance.

	Original		remove 10%		remove 15%		remove 20%		remove 25%		remove 30%		remove 35%		remove 40%		remove 45%		remove 50%		remove 55%		remove 60%	
abs(PCC)>0.5, FDR cut off	gene-peak links	overlapped	gene-peak links	overlapped	gene-peak links	overlapped	gene-peak links	overlapped	gene-peak links	overlapped	gene-peak links	overlapped	gene-peak links	overlapped	gene-peak links	overlapped	gene-peak links	overlapped	gene-peak links	overlapped	gene-peak links	overlapped	gene-peak links	overlapped
0.05	707	498	693	542	684	572	614	575	591	591	544	544	507	507	474	474	476	476	432	432	410	410	313	313
0.1	1,234	853	1,203	938	1,193	988	1,088	1,005	1,082	1,082	984	984	910	910	832	832	828	828	726	726	661	661	523	523
0.2	2,567	1,676	2,492	1,854	2,403	1,950	2,197	1,982	2,085	2,085	1,950	1,950	1,805	1,805	1,579	1,579	1,424	1,424	1,212	1,212	1,030	1,030	806	806
0.3	3,515	2,257	3,036	2,257	2,798	2,257	2,508	2,257	2,257	2,257	2,031	2,031	1,805	1,805	1,579	1,579	1,424	1,424	1,212	1,212	1,030	1,030	806	806
0.4	3,515	2,257	3,036	2,257	2,798	2,257	2,508	2,257	2,257	2,257	2,031	2,031	1,805	1,805	1,579	1,579	1,424	1,424	1,212	1,212	1,030	1,030	806	806
0.5	3,515	2,257	3,036	2,257	2,798	2,257	2,508	2,257	2,257	2,257	2,031	2,031	1,805	1,805	1,579	1,579	1,424	1,424	1,212	1,212	1,030	1,030	806	806
0.6	3,515	2,257	3,036	2,257	2,798	2,257	2,508	2,257	2,257	2,257	2,031	2,031	1,805	1,805	1,579	1,579	1,424	1,424	1,212	1,212	1,030	1,030	806	806
0.7	3,515	2,257	3,036	2,257	2,798	2,257	2,508	2,257	2,257	2,257	2,031	2,031	1,805	1,805	1,579	1,579	1,424	1,424	1,212	1,212	1,030	1,030	806	806
0.8	3,515	2,257	3,036	2,257	2,798	2,257	2,508	2,257	2,257	2,257	2,031	2,031	1,805	1,805	1,579	1,579	1,424	1,424	1,212	1,212	1,030	1,030	806	806
0.9	3,515	2,257	3,036	2,257	2,798	2,257	2,508	2,257	2,257	2,257	2,031	2,031	1,805	1,805	1,579	1,579	1,424	1,424	1,212	1,212	1,030	1,030	806	806
# all pairs	1,019,530		813,651		720,826		635,666		555,409		478,822		410,616		344,534		285,061		230,314		182,196		140,076	

Point 2: Same concern is noted for Major 4 as for Major 3 (see above).

We thank Reviewer #3 for raising this important concern. For the peak-drug linkage, we also follow the Reviewer #3's suggestions and revise our procedure. In this study, there are 64 drugs used for drug-peak correlation analysis and the total number of drug-peak pairs is 10,030,144. Since acknowledged biologically informative co-variants such as genomic distance is not available for drug-peak correlation

analysis. To increase the detection power, we try to filter the drug-peak pairs by removing the bottom 10% to 60% of peaks with low variance of accessibility but keep all drugs due to its relatively small cohort. The results summarized in the following table (Referee table 2) indicate that this filtering could only slightly reduce the testing drug-peak pairs from 10 million to 4 million and have limited impact on controlling the false discoveries. Furthermore, we tried different ways to transform the raw input data to improve the analysis. As shown in the following table (Referee table 3), this exponential transformation generates more drug-gene linkages when using the same FDR cut-off such as 0.6 and 0.7 with absolute PCC>0.5. For example, there is 1 pair with FDR<0.6 and absolute PCC>0.5 under removal of the bottom 30% of peaks with low variance, and this number increased to 2,181 pairs after exponential transformation. However, vast majority of their FDR is still >0.5. Therefore, in the revised manuscript, we finally choose to use absolute PCC>0.5 and raw p-value<0.01 to identify the drug-peak linkages, which focus more on single test and is also a common strategy to identify linkages in high-throughput experiment (Shen et al., Nature, 2012, PMID: 22763441).

Referee table 2. The number of identified peak-drug linkages and their overlap with the identified 15,397 linkages based on different cut-offs of FDR and absolute PCC>0.5 under removing the bottom 10%-60% of peaks with low variance.

	Original		remove 10%		remove 15%		remove 20%		remove 25%		remove 30%		remove 35%		remove 40%		remove 45%		remove 50%		remove 55%		remove 60%		
	gene-peak links	overlapped	gene-peak links	overlapped	gene-peak links	overlapped	gene-peak links	overlapped	gene-peak links	overlapped	gene-peak links	overlapped	gene-peak links	overlapped	gene-peak links	overlapped	gene-peak links	overlapped	gene-peak links	overlapped	gene-peak links	overlapped	gene-peak links	overlapped	
abs(PCC)>0.5, FDR cut off																									
0.05	1	1	1	1	1	1	1	1	1	1	1	1	1	1	1	1	1	1	1	1	1	1	1	1	
0.1	1	1	1	1	1	1	1	1	1	1	1	1	1	1	1	1	1	1	1	1	1	1	1	1	
0.2	1	1	1	1	1	1	1	1	1	1	1	1	1	1	1	1	1	1	1	1	1	1	1	1	
0.3	1	1	1	1	1	1	1	1	1	1	1	1	1	1	1	1	1	1	1	1	1	1	1	1	
0.4	1	1	1	1	1	1	1	1	1	1	1	1	1	1	1	1	1	1	1	1	1	1	1	1	
0.5	1	1	1	1	1	1	1	1	1	1	1	1	1	1	1	1	1	1	1	1	1	1	1	1	
0.6	1	1	1	1	1	1	1	1	1	1	1	1	1	1	1	1	1	1	1	1	1	1	1	1	
0.7	1	1	1	1	1	1	8	8	1	1	7	7	1	1	6	6	1	1	5	5	1	1	2	2	
0.8	15,397	15,397	13,981	13,981	13,242	13,242	12,568	12,568	11,796	11,796	10,974	10,974	10,312	10,312	9,636	9,636	8,821	8,821	8,113	8,113	7,258	7,258	6,376	6,376	
0.9	15,397	15,397	13,981	13,981	13,242	13,242	12,568	12,568	11,796	11,796	10,974	10,974	10,312	10,312	9,636	9,636	8,821	8,821	8,113	8,113	7,258	7,258	6,376	6,376	
# all pairs	10,030,144		9,027,328		8,525,824		8,024,320		7,522,816		7,021,248		6,519,744		6,018,240		5,516,736		5,015,232		4,513,664		4,012,160		

Referee table 3. The number of identified peak-drug linkages and their overlap with the identified 15,397 linkages based on different cut-offs of FDR and absolute PCC>0.5 under removing the bottom 10%-60% of peaks with low variance after exponential transformation.

	Original		remove 10%		remove 15%		remove 20%		remove 25%		remove 30%		remove 35%		remove 40%		remove 45%		remove 50%		remove 55%		remove 60%	
	gene-peak links	overlapped	gene-peak links	overlapped	gene-peak links	overlapped	gene-peak links	overlapped	gene-peak links	overlapped	gene-peak links	overlapped	gene-peak links	overlapped	gene-peak links	overlapped	gene-peak links	overlapped	gene-peak links	overlapped	gene-peak links	overlapped	gene-peak links	overlapped
abs(PCC)>0.5, FDR cut off																								
0.05	0	0	0	0	0	0	0	0	0	0	0	0	0	0	0	0	0	0	0	0	0	0	0	0
0.1	0	0	0	0	0	0	0	0	0	0	0	0	0	0	0	0	0	0	0	0	0	0	0	0
0.2	0	0	0	0	0	0	0	0	0	0	0	0	1	1	1	1	1	1	1	1	0	0	0	0
0.3	1	1	1	1	1	1	1	1	1	1	1	1	1	1	1	1	1	1	1	1	0	0	0	0
0.4	1	1	1	1	1	1	1	1	1	1	1	1	2	2	2	2	1	1	1	1	0	0	0	0
0.5	1	1	1	1	1	1	23	23	2	2	2	2	19	19	2	2	1	1	1	1	0	0	0	0
0.6	643	632	668	654	621	607	2,674	2,456	2,711	2,463	2,181	1,988	8,117	5,451	11,959	6,562	11,105	6,004	10,216	5,495	9,241	4,884	8,244	4,267
0.7	17,924	10,866	16,522	9,741	15,767	9,183	15,082	8,694	14,292	8,128	13,408	7,506	12,717	7,039	11,959	6,562	11,105	6,004	10,216	5,495	9,241	4,884	8,244	4,267
0.8	17,924	10,866	16,522	9,741	15,767	9,183	15,082	8,694	14,292	8,128	13,408	7,506	12,717	7,039	11,959	6,562	11,105	6,004	10,216	5,495	9,241	4,884	8,244	4,267
0.9	17,924	10,866	16,522	9,741	15,767	9,183	15,082	8,694	14,292	8,128	13,408	7,506	12,717	7,039	11,959	6,562	11,105	6,004	10,216	5,495	9,241	4,884	8,244	4,267
# all pairs	10,030,144		9,027,328		8,525,824		8,024,320		7,522,816		7,021,248		6,519,744		6,018,240		5,516,736		5,015,232		4,513,664		4,012,160	

Comparing to above gene-peak pairs, we realize that the FDR adjustment for drug-peak pairs is far more

challenging. We summarize the reason as follows. First, the total number of the tests is large and we are lack of informative co-variant to efficiently reduce the number tests as genome distance does in gene-peak case. Thus, finding biologically informative co-variants such as genomic distance is in pressing need to efficiently narrow down the total number of drug-peak linkages. Second, due to relatively small cohort of drug, stringent variance filtering is only performed on peaks with low variance of accessibility value, which is not powerful enough to control the false discoveries. We have cautiously interpreted the 15,397 drug-peak linkages in the revised manuscript, and more validations are needed to support those candidate pairs in the future. We thank the Reviewer #3 again to clear our thoughts regarding to this limitation.

Point 3: *The authors have responded adequately to Major 5. The authors should clarify in the manuscript that the 10 representative examples used were picked to show the most extreme differences. Referee Fig 9 is very encouraging; the authors could consider including this in Supplementary.*

We thank Reviewer #3 for the valuable advice and have clarified that 10 representative examples were selected to show the most extreme differences in Fig. 5k-m and Extended Data Fig. 5g-q in the revised manuscript. Moreover, we agree with Reviewer #3 that Referee Fig 9 provided very encouraging results and have added it into the revised manuscript as Extended Data Fig. 6.

Point 4: *The authors have indicated in the manuscript that the somatic mutational profiles of the PDCPOs were similar to that observed in the TCGA pancreatic adenocarcinoma (PAAD) dataset. To back this claim, the authors should add Referee Fig 12 to Supplementary.*

We thank Reviewer #3 for the constructive suggestion to help improve our study. Following this advice, we have added Referee Fig 12 into the revised manuscript as Extended Data Fig. 2b to support the results about that the somatic mutational profiles of the PDCPOs were similar to that observed in the TCGA pancreatic adenocarcinoma (PAAD) dataset.

REVIEWERS' COMMENTS

Reviewer #3 (Remarks to the Author):

The authors addressed all points that I raised in the previous review.

Point-by-point response to Reviewers' comments

Dec 16th, 2021

RE: NCOMMS-21-10013B

Integrated profiling of human pancreatic cancer organoids reveals chromatin accessibility to drug sensitivity networks

Summary statement. We are very grateful to the Editor for handling our manuscript. We also appreciate very much for Reviewers for their constructive suggestions and positive comments on our manuscript.

Reviewer #1

The authors addressed all points that I raised in the previous review.

We thank Reviewer #1 for raising positive comments and providing valuable suggestions.

Reviewer #2

The authors have addressed my concerns.

We thank Reviewer #2 for the positive comments and raising helpful advice.

Reviewer #3

The authors addressed all points that I raised in the previous review.

We are grateful to Reviewer #3 for the positive comments and help in improving our manuscript.